# Unanchored tri-NEDD8 inhibits PARP-1 to protect from oxidative stress-induced cell death

Matthew J Keuss[1], Roland Hjerpe[1], Oliver Hsia[1], Robert Gourlay[2], Richard Burchmore[3], Matthias Trost[2,4] & Thimo Kurz[1,*]

## Abstract

NEDD8 is a ubiquitin-like protein that activates cullin-RING E3 ubiquitin ligases (CRLs). Here, we identify a novel role for NEDD8 in regulating the activity of poly(ADP-ribose) polymerase 1 (PARP-1) in response to oxidative stress. We show that treatment of cells with $H_2O_2$ results in the accumulation of NEDD8 chains, likely by directly inhibiting the deneddylase NEDP1. One chain type, an unanchored NEDD8 trimer, specifically bound to the second zinc finger domain of PARP-1 and attenuated its activation. In cells in which *Nedp1* is deleted, large amounts of tri-NEDD8 constitutively form, resulting in inhibition of PARP-1 and protection from PARP-1-dependent cell death. Surprisingly, these NEDD8 trimers are additionally acetylated, as shown by mass spectrometry analysis, and their binding to PARP-1 is reduced by the overexpression of histone de-acetylases, which rescues PARP-1 activation. Our data suggest that trimeric, acetylated NEDD8 attenuates PARP-1 activation after oxidative stress, likely to delay the initiation of PARP-1-dependent cell death.

**Keywords** cell death; NEDD8; oxidative stress; PARP-1; parthanatos
**Subject Categories** Autophagy & Cell Death; Post-translational Modifications, Proteolysis & Proteomics
**The EMBO Journal (2019) 38: e100024**

## Introduction

Many cellular processes are regulated via the post-translational modification of proteins with ubiquitin and ubiquitin-like proteins (UBLs). UBLs are small proteins of approximately 8 kDa molecular mass that become covalently linked to other proteins via an isopeptide bond, usually on lysine residues (van der Veen & Ploegh, 2012). The conjugation of ubiquitin/UBLs to a substrate entails three enzymatic steps. First, an E1 activating enzyme adenylates ubiquitin/UBL at its C-terminus and then links ubiquitin/UBL to the enzyme's active-site cysteine via a thioester bond (Ciechanover

et al, 1981). Next, the ubiquitin/UBL is passed from E1 to the active-site cysteine of an E2-conjugating enzyme (Hershko et al, 1983). In the final step, this ubiquitin/UBL-charged E2 is recruited by E3 ligases, which also bind to the substrate (Hershko et al, 1983). Ubiquitin/UBL is then transferred onto a ε-amino group of a lysine residue of the substrate, where it forms an isopeptide bond (Goldknopf & Busch, 1977; Hershko et al, 1983; Scheffner et al, 1995). After the initial linkage of the first UBL to its substrate, many ubiquitin/UBLs then form lysine-linked chains on their substrates to regulate diverse cellular processes (Yau & Rape, 2016). Poly-ubiquitylation with a lysine-48-linked ubiquitin chain, for example, results in the proteasomal degradation of the substrate (Chau et al, 1989). Free, non-conjugated chains of ubiquitin have also been described and play an important role in the immune response (Zeng et al, 2010). Although many similarities exist between ubiquitin and ubiquitin-like modifiers, free, unanchored chains have not been similarly described for UBLs.

NEDD8 is the ubiquitin-like modifier that is most similar to ubiquitin, and its best characterized function is the regulation of a class of enzymes that ligate ubiquitin to its substrates (Enchev et al, 2015). These E3 ubiquitin ligase enzymes are multi-subunit complexes that are built around a core scaffolding cullin protein [cullin-RING E3 ubiquitin ligases (CRLs)]; the mono-neddylation of the cullin subunit activates CRLs (Morimoto et al, 2000; Read et al, 2000; Ohh et al, 2002; Petroski & Deshaies, 2005; Enchev et al, 2015). Numerous non-cullin substrates of NEDD8 have also been reported, but their physiological importance has been called into doubt by the realization that NEDD8 can enter the ubiquitin pathway when overexpressed (Hjerpe et al, 2012a,b). Thus, careful studies with endogenous NEDD8 are needed to confirm that it indeed has non-cullin substrates (Enchev et al, 2015).

There is indirect evidence to suggest that NEDD8 functions independently of its role in cullin activation. Most eukaryotic genomes encode two very specific de-neddylating enzymes, the COP9 signalosome, which only de-neddylates cullins, and the deneddylase NEDP1, which cannot cleave NEDD8 from cullins, but which appears to be promiscuous in its ability to de-neddylate other proteins, including those erroneously neddylated by ubiquitin enzymes (Gan-Erdene et al, 2003; Mendoza et al, 2003;

1 Henry Wellcome Lab of Cell Biology, College of Medical, Veterinary and Life Sciences, Institute of Molecular, Cell and Systems Biology, University of Glasgow, Glasgow, UK
2 The MRC Protein Phosphorylation and Ubiquitylation Unit, The Sir James Black Centre, College of Life Sciences, University of Dundee, Dundee, UK
3 Glasgow Polyomics, College of Veterinary, Medical and Life Sciences, University of Glasgow, Glasgow, UK
4 Institute for Cell and Molecular Biosciences, Newcastle University, Newcastle upon Tyne, UK
*Corresponding author. Tel: +44 141 330 3908; E-mail: thimo.kurz@glasgow.ac.uk

Wu *et al*, 2003; Watson *et al*, 2006; Broemer *et al*, 2010). The physiological substrates of NEDP1, however, remain largely unknown. Furthermore, poly-NEDD8 chains have been identified by mass spectrometry, and as cullins are only mono-neddylated, this finding strongly suggests that NEDD8 fulfils other roles (Kim *et al*, 2011; Leidecker *et al*, 2012; Coleman *et al*, 2017). Similarly, the overall cellular neddylation pattern of proteins changes when cells are exposed to oxidative stress (Leidecker *et al*, 2012). However, the NEDD8 substrates that respond to oxidative stress also remain unknown (Leidecker *et al*, 2012). Increased levels of neddylated proteins have also been detected in flies, mice and cell lines in which the endogenous NEDP1-encoding gene has been genetically inactivated (Chan *et al*, 2008; Vogl *et al*, 2015; Coleman *et al*, 2017), but these molecules have not been functionally characterized or identified. While recent work in *Arabidopsis* and in mammalian cells has demonstrated that NEDP1 de-neddylates components of the NEDD8 conjugation machinery (Mergner *et al*, 2015, 2017; Coleman *et al*, 2017), these findings cannot fully account for the increased neddylation observed in knockout cells.

In this study, we report that cells efficiently generate non-cullin conjugates in the form of poly-NEDD8 chains. Normally, these chains do not accumulate due to their cleavage by NEDP1, but our findings show that NEDD8 chains are stabilized when NEDP1 is inhibited, either genetically or through oxidative stress. We discovered that one type of NEDD8 chain, an unanchored, trimeric NEDD8, interacts with poly(ADP-ribose) polymerase 1 (PARP-1) and attenuates its activation. PARP-1 overactivation mediates cell death after oxidative stress, and our data suggest that tri-NEDD8 prevents PARP-1 hyperactivation to delay premature commitment to cell death.

## Results

To better understand the non-cullin roles of NEDD8, we generated U2OS- and HEK 293-NEDP1 knockout cell lines using the CRISPR/Cas9 system (Cong *et al*, 2013; Mali *et al*, 2013). Consistent with previous work, the genetic deletion of *Nedp1* led to the accumulation of neddylated species that do not migrate at the ~ 100 kDa size of neddylated cullins in both cell lines (Figs 1A and EV1A). Interestingly, the NEDD8 reactive bands were spaced very evenly and were distributed throughout the molecular mass range of the gel. The bands started at ~ 15 kDa, which corresponds in size to a NEDD8 dimer, and ranged in size up to high molecular mass bands of > 130 kDa (Fig 1A). The abundance of neddylated proteins was so high following the genetic deletion of *Nedp1* that non-conjugated free NEDD8 was depleted, indicating that these conjugates formed and accumulated efficiently in the absence of NEDP1 (Figs 1A and EV1A).

To further characterize these neddylated species, we next purified them from NEDP1 knockout cells using a NEDD8 affinity resin we generated by fusing the HALO protein with a catalytically inactive form of NEDP1, in which the active-site cysteine was mutated to alanine (C163A). When exposed to cell extract, this resin efficiently enriched neddylated proteins from both NEDP1 KO cells and from WT cells, but it did not enrich for ubiquitylated proteins (Fig 1B). The enrichment from WT cell extracts revealed the same

non-cullin, neddylated species to be present in WT cells, but they were below the detection limit of Western blot analysis without prior enrichment (Fig 1B). Further mutation of NEDP1 at key residues D29W, A98K and G99K, previously identified in the NEDP1-NEDD8 co-crystal structure (Shen *et al*, 2005), led to a significant reduction in the ability of NEDP1 to bind NEDD8 (Fig 1C), which allowed us to utilize this form of the protein as a negative control in pulldown experiments.

Previous reports have demonstrated that the enzymes of the NEDD8 pathway are neddylated when NEDP1 is deleted in *Arabidopsis* and in mammalian cells (Mergner *et al*, 2015, 2017; Coleman *et al*, 2017), which we could confirm in our NEDP1 KO cells. In these cells, both subunits of the NEDD8 E1 enzyme (ULA1/UBA3) and the NEDD8 E2 enzyme UBE2M, as well as the NEDD8 co-E3s DCNL1 and DCNL2, were neddylated (Fig 1C). We also co-purified hyper-neddylated forms of Cul2 and Cul3 from NEDP1 KO cells (Fig 1C). This is consistent with data showing that NEDP1 can deconjugate NEDD8 from hyper-neddylated cullins *in vitro* (Wu *et al*, 2003). These results support previous work that has concluded that NEDP1 cleaves NEDD8 from the enzymes involved in cullin neddylation, as well as from erroneously neddylated cullins (Mergner *et al*, 2015; Coleman *et al*, 2017). Although we were able to detect the neddylated forms of the NEDD8 conjugation machinery after enrichment, they were not overly abundant in whole-cell lysates (Fig 1D), suggesting that the majority of neddylation events in NEDP1 KO cells do not correspond to these enzymes. Instead, we speculated that given their even ~ 8-kDa spacing, the NEDD8 reactive bands might instead represent NEDD8-NEDD8 linkages and, possibly, unanchored NEDD8 chains (Fig 1A).

NEDD8 chain formation has been previously reported to occur *in vitro* (Ohki *et al*, 2009); we were also able to generate unanchored NEDD8 chains *in vitro* using recombinant NEDD8 E1 enzyme (NAE) and the NEDD8 E2 enzyme, UBE2M, but in the absence of any NEDD8 E3. The smallest protein in these *in vitro* reactions, aside from NEDD8 (8.5 kDa), was UBE2M, with a molecular mass of ~ 22 kDa. The smallest possible, non-NEDD8-NEDD8 conjugate in these reactions, a UBE2M-NEDD8 linkage, would thus have a molecular mass of ~ 31 kDa. However, while this product was present in the reaction, conjugates were also formed at lower molecular masses that could only correspond to unanchored NEDD8 dimers, trimers and tetramers (Fig 1E). Mass spectrometry analysis of the *in vitro* reactions revealed that NEDD8 chains were indeed formed and conjugated to lysine residues 4, 6, 11, 22, 27, 48, 54 and 60 (Fig 1F).

The migration pattern of unanchored NEDD8 chains generated *in vitro* looked very similar to the lower molecular mass NEDD8 reactive species apparent in NEDP1 knockout cells (Fig 1A and E). To determine whether these are indeed similar to the NEDD8 chains generated *in vitro*, we next focused on the pathways that form the non-cullin neddylated species in cells. We found that non-cullin neddylation in NEDP1 knockout cells was generated by bona fide NEDD8 enzymes, and not by ubiquitin enzymes, as evidenced by the reduction in NEDD8 reactive species following treatment with the NEDD8 E1 enzyme (NAE) inhibitor (MLN4924) alone, and not following treatment with the ubiquitin E1 enzyme (UBE) inhibitor (MLN7243) (Bahamon *et al*, 2014; Brownell *et al*, 2010; Soucy *et al*, 2009; Figs 1G and EV1B). Furthermore, siRNA-mediated knockdown of only one of the two NEDD8 E2 enzymes, UBE2M, but not

    

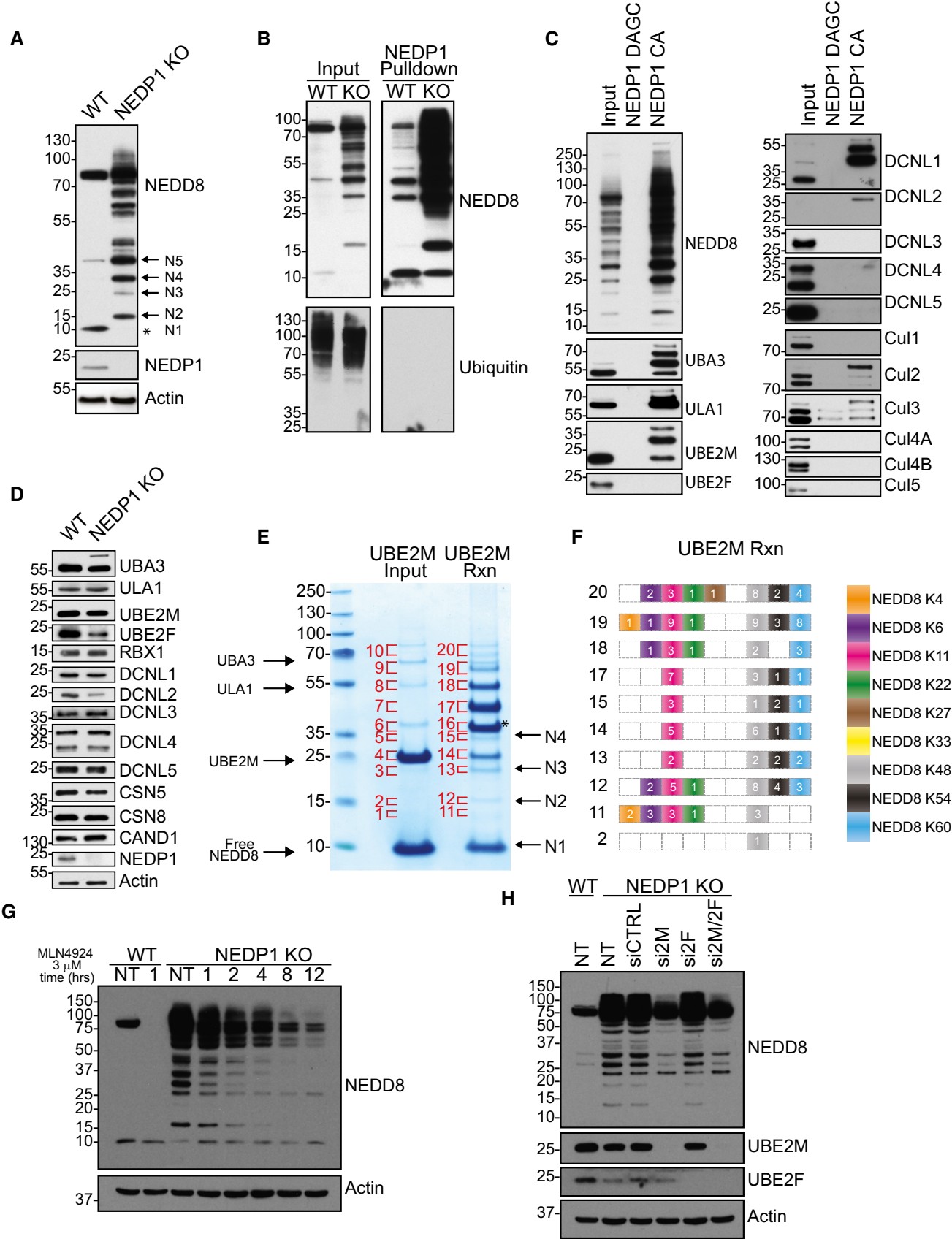

**Figure 1.**

◄

**Figure 1. Generation and analysis of NEDP1 knockout HEK 293 cells.**

A Western blot analysis of whole-cell lysates from HEK 293 WT and NEDP1 KO cells reveals a loss of free NEDD8 (indicated by asterisk) and an accumulation of NEDD8 reactive species in the NEDP1 KO lysate. The predicted molecular weight sizes of putative, unanchored, poly-NEDD8 chains are denoted by N2 through to N5. Unconjugated NEDD8 is denoted by N1.

B NEDD8 affinity resin shows enrichment of endogenous neddylated proteins in WT and NEDP1 KO cells. Recombinant HALO-NEDP1 C163A (CA) conjugated to HALO-Link beads was used as an affinity resin to enrich for neddylated proteins in lysates from HEK 293 WT and NEDP1 KO cells. Enriched proteins were resolved by SDS–PAGE and processed for Western blot analysis with NEDD8 or ubiquitin antibodies. HALO-NEDP1 CA specifically enriches for NEDD8-reactive proteins in both WT and NEDP1 KO cells, but does not enrich for Ubiquitin-modified proteins in either cell line.

C Components of the NEDD8 conjugation machinery are enriched in HALO-NEDP1 pulldowns from NEDP1 KO lysates. Neddylated proteins from HEK 293 KO cells were enriched by HALO-NEDP1 CA pulldown, as in (B) but not by the NEDD8 nonbinder mutant, HALO-NEDP1 DAGC (D29W A98K G99K C163A). The NEDD8 E1s, UBA3 and ULA1, are modified in NEDP1 KO cells, as well as E2 UBE2M, and co-E3s DCNL1 and DCNL2. Cul2 and Cul3 are hyper-neddylated in NEDP1 KO cells. CSN components, CSN5 and CSN8, also co-precipitate in HALO-NEDP1 CA pulldowns.

D Western blot analysis from HEK 293 WT and NEDP1 KO cells of the components of the NEDD8 conjugation/de-conjugation pathway shows that similar levels of NEDD8 pathway components are present in both WT and NEDP1 KO cells. Apart from UBA3, there is no detectable amount of NEDD8-modified enzymes in whole-cell lysates from NEDP1 KO cells.

E Poly-NEDD8 chains can be generated by *in vitro* reactions (Rxn). NAE (0.15 μM), UBE2M and NEDD8 (20 μM) were incubated on ice or incubated at 30°C for 3 h and reactions were stopped by addition of LDS sample loading buffer. Reactions were resolved by SDS–PAGE and stained with colloidal Coomassie. Indicated bands were excised from the gel and processed for in-gel trypsin digestion and mass spectrometry analysis. The predicted molecular weight sizes for a theoretical unanchored NEDD8 chain are denoted by N2-N4. Unconjugated NEDD8 is indicated by N1. UBE2M modified by NEDD8 is indicated with an asterisk.

F Diagram of the NEDD8 linkages, as determined by mass spectrometry analysis, from (E), with the number of spectral counts indicated for the bands labelled in (E). Only bands with identified diGly motifs are shown here. UBE2M generates *in vitro* chains of poly-NEDD8 with linkages on K4, K6, K11, K22, K27, K48, K54 and K60.

G Neddylated species are NEDD8 E1 dependent. WT and NEDP1 KO HEK 293 cells were treated with NAE inhibitor MLN4924 at 3 μM for the indicated time. Lysed cells were then processed for Western blot analysis. NEDD8 E1 inhibition results in a time-dependent decrease in the amount of Cullin and non-Cullin NEDD8 reactive bands.

H Neddylated species are UBE2M dependent. WT and NEDP1 KO HEK 293 cells were left untreated or treated with the indicated siRNA for 48 h. Lysed cells were then processed for Western blot analysis. Neddylated species are reduced when NEDD8 E2 UBE2M was depleted after siRNA treatment (si2M), but not with control siRNA (siCTRL) or when NEDD8 E2 UBE2F was depleted (si2F). The double knockdown (si2M/2F) does not further reduce NEDD8 modified proteins.

of UBE2F, reduced non-cullin neddylation in NEDP1 knockout cells, demonstrating that their formation is UBE2M dependent (Fig 1H). Knockdown of the NEDD8 co-E3s DCUN1D1-5/DCNL1-5/SCCRO1-5 (Kurz *et al*, 2005; Keuss *et al*, 2016), which aid in cullin neddylation, had no effect on the neddylated species (Fig EV1C), suggesting that the DCNLs are either redundant or that the neddylated species are formed by other mechanisms, possibly independently of an E3. Therefore, as with the *in vitro* reactions that generate NEDD8 chains, the NEDD8 conjugates in NEDP1 KO cells are also generated by NAE1 and UBE2M. Furthermore, many of the bands in NEDP1 knockout cells migrate to multiples of ~ 8 kDa, which indicates that they might indeed represent free, unanchored NEDD8 chains (Fig 1A). The "smallest" possible NEDD8 chain would be a NEDD8-NEDD8 dimer, migrating at ~ 17 kDa. In NEDP1 knockout cells, a putative dimeric form of NEDD8 was indeed clearly visible, as were further multiples of up to a tetramer (Fig 1A). This further supports the notion that at least some of the conjugates in NEDP1 KO cells represent unanchored NEDD8 chains of varying lengths (Fig 1A).

To confirm the existence of NEDD8 chains in cells, we next analysed large-scale enrichments of neddylated proteins from HEK 293 NEDP1 KO lysates by mass spectrometry, following in-gel trypsin digest (Fig 2A). In particular, we analysed the di-glycine (diGly) remnants that remain on lysine residues following the trypsin digestion of neddylated proteins. Modification with ubiquitin leaves an identical diGly remnant; however, as shown by Western blot, our purifications do not contain ubiquitin (Fig 1B). This also excludes the possibility of mixed NEDD8/ubiquitin conjugates being present in the purifications. Using a cut-off of three identified diGly-containing peptides to increase confidence in our results, we only detected two modified proteins: the methyl-transferase NSUN2, with four identified diGly peptides, and NEDD8 itself, with a total of 44 identified diGly peptides from K11, K48 and K54 modifications, confirming the existence of NEDD8 chains in NEDP1 knockout cells

(Fig 2B). We were unable to verify the neddylation of NSUN2 in WT or NEDP1 KO cells when we overexpressed FLAG-NSUN2 followed by anti-FLAG immunoprecipitation (Fig EV2), indicating that this neddylation may not take place on overexpressed and tagged NSUN2 or that it may be of very low abundance. This suggests that most neddylation events occur on NEDD8 itself. To confirm this assumption, we next analysed the mass spectrometry data using label-free quantitation by MaxQuant combined with intensity-based, absolute quantification iBAQ analysis (Fig 2C; Cox & Mann, 2008; Cox *et al*, 2011; Silva *et al*, 2006). iBAQ quantifies protein abundance by taking the length of the protein and the number of theoretically detectable peptides into account. This was important, as, at 8.5 kDa, NEDD8 is much smaller than most proteins identified in the sample. From this analysis, we were able to conclude that NEDD8 is over ten times more abundant in our preparations than is the second most abundant protein, the NEDD8 E2 UBE2M, and that NEDD8 constitutes ~ 75% of all protein content identified (Fig 2C). Thus, NEDD8 is highly enriched in our purifications, which strongly suggests that NEDD8 forms either free chains or very long chains on putative non-NEDD8 substrates. However, given that, with the exception of NSUN2, by mass spectrometry we only detect diGly motifs on NEDD8 itself, even in gel slices cut from low molecular mass ranges (which could otherwise only correspond to a single NEDD8 moiety ligated to another small protein), this strongly suggests that NEDD8 indeed forms unanchored chains in cells.

To better understand the physiological role of NEDD8 chains, we next examined the mass spectrometry data in greater detail for proteins that co-purified with poly-NEDD8. From this analysis, we identified components of the neddylation pathway, but also a striking enrichment of the proteins known to be modified by poly (ADP-ribosyl)ation (PAR), including the poly(ADP-ribose) polymerase PARP-1 (Fig 2C; Table EV1). In fact, 178 of the 692 proteins

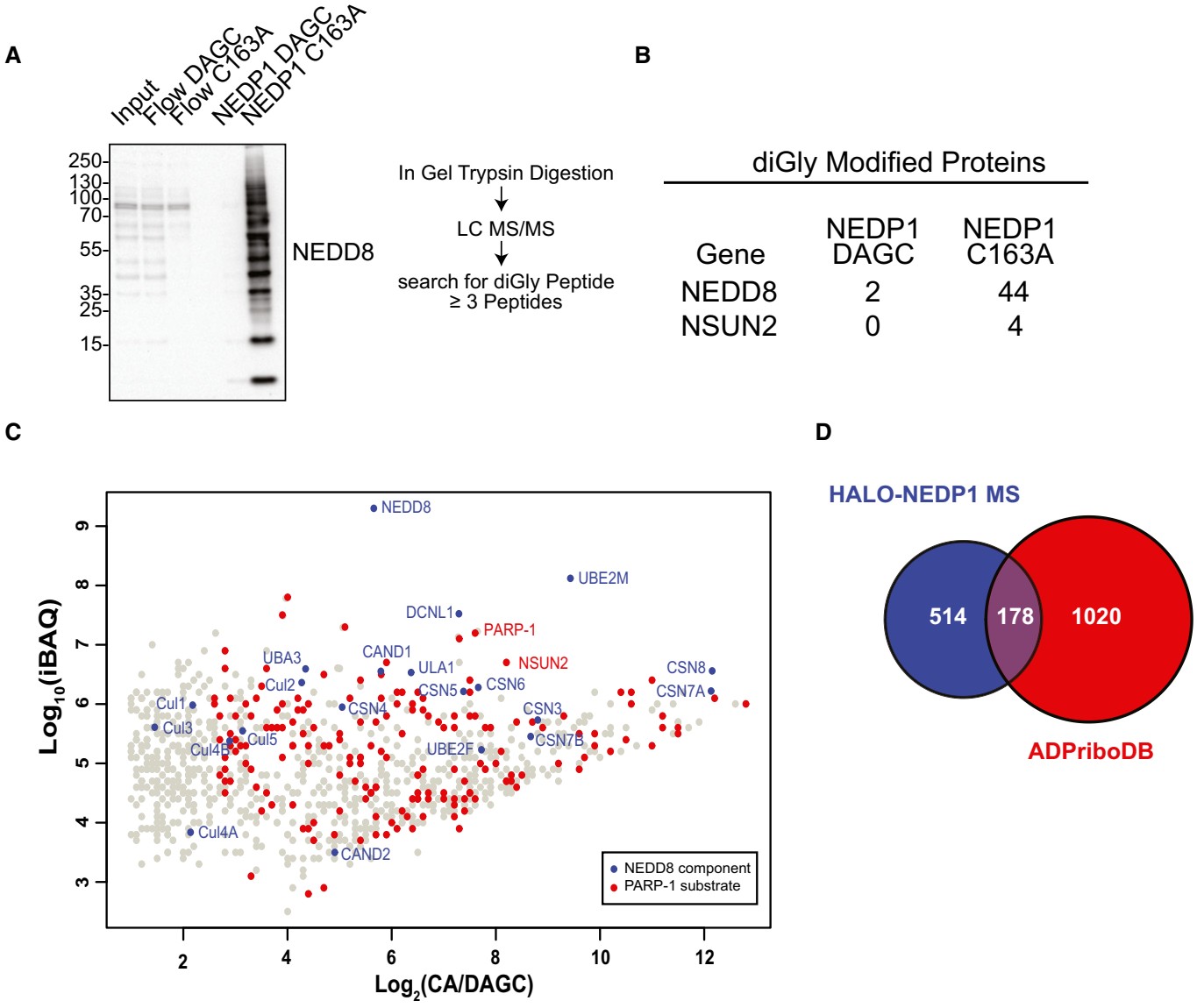

**Figure 2. NEDP1 knockout results in the accumulation of poly-NEDD8.**

A   (Left) Western blot analysis of large-scale HALO-NEDP1 pulldowns from HEK 293 NEDP1 KO lysates. (Right) Diagram of mass spectrometry sample preparation.

B   Table of the number of spectral counts from proteins with identified diGly-modified peptides, following pulldown by HALO-NEDP1 CA or the HALO-NEDP1 DAGC reduced binder control. NEDD8 is the protein most highly modified by NEDD8.

C   Scatter plot of proteins identified by mass spectrometry analysis identifies NEDD8 and components of the NEDD8 conjugation pathway as being the most abundant proteins in HEK 293 NEDP1 KO lysates following NEDP1-CA pulldown. iBAQ analysis of proteins identified in (A) is plotted as the $\log_2$ value of the enrichment ratio (mass spectrometry intensity of the HALO-NEDP1 CA pulldown over HALO-NEDP1 DAGC pulldown) versus the $\log_{10}$ value of the iBAQ intensity from the HALO-NEDP1 CA pulldown. Blue markers indicate known components of the NEDD8 pathway, and red markers indicate proteins that have been identified as substrates of PARP-1 in the database of ADP-ribosylated proteins, ADPriboDB (Vivelo *et al*, 2017).

D   Venn diagram of the proteins enriched by at least sixfold following HALO-NEDP1 CA pulldown compared with proteins identified as PARP-1 substrates in the database of ADP-ribosylated proteins, ADPriboDB (Vivelo *et al*, 2017).

enriched by HALO-NEDP1 pulldown have previously been identified as being ADP-ribosylated proteins (Fig 2D; Vivelo *et al*, 2017).

PARP-1 and poly(ADP-ribosyl)ation are known regulators of DNA repair and of the oxidative stress response. PARP-1 senses DNA damage caused by oxygen radicals and aids in the early steps of DNA repair (El-Khamisy *et al*, 2003; Le Page *et al*, 2003). PARP-1 consumes nicotinamide adenine dinucleotide ($NAD^+$) to covalently

attach ADP-ribose to various amino acids, including aspartic acid, glutamic acid, lysine, arginine and serine, that are present on itself and on its substrate proteins (Daniels *et al*, 2014; Leidecker *et al*, 2016). PARP-1 contains three zinc-binding (Zn) domains in its N-terminus, a breast cancer one protein C-terminal (BRCT) domain, and a tryptophan-, glycine-, arginine-rich (WGR) domain at its centre, as well as a C-terminal catalytic domain (Altmeyer *et al*,

2009; Langelier *et al*, 2012). Upon binding to damaged DNA via its Zn fingers, the catalytic activity of PARP-1 increases dramatically from a low basal level (Rouleau *et al*, 2010). PARP-1-dependent poly(ADP-ribosyl)ation at sites of DNA damage is an important step in the timely repair of DNA, but it is also implicated in inducing cell death after excessive DNA damage (Durkacz *et al*, 1980; Berger, 1985; Satoh & Lindahl, 1992; Ha & Snyder, 1999). When strongly activated, PARP-1 depletes the cytosolic pool of $NAD^+$ and generates long-chain, branched PAR polymer which accumulates in cells (Zong *et al*, 2004). Both of these events have been linked to the release of apoptosis-inducing factor (AIF) from mitochondria and to the activation of cell death (Andrabi *et al*, 2006; Alano *et al*, 2010). $NAD^+$ depletion blocks glycolysis at the GAPDH step, which leads to a loss of carbon sources for mitochondria and to the subsequent reduction of ATP and to mitochondrial depolarization, resulting in a programmed form of necrotic cell death (Alano *et al*, 2010). The direct binding of PAR polymer to AIF is sufficient to release AIF from mitochondria and to activate the cell death pathway; a process termed parthanatos (Andrabi *et al*, 2006; David *et al*, 2009; Wang *et al*, 2011). Once released from mitochondria, AIF translocates to the nucleus and binds macrophage migration inhibitory factor (MIF), which induces MIF's nuclease activity, resulting in large-scale DNA fragmentation (Wang *et al*, 2016).

Oxidative stress is one of the strongest inducers of PARP-1 and has been previously shown to generate non-cullin conjugates of NEDD8 (Leidecker *et al*, 2012). Indeed, our findings show that when cells are exposed to increasing levels of oxidative stress in the form of $H_2O_2$, the quantity of NEDD8 reactive bands detected by Western blot significantly and rapidly increased across a range of sizes (Fig 3A). The most prominently generated NEDD8 conjugate produced in response to oxidative stress migrated at ~ 25 kDa. As NEDD8 itself is 8.5 kDa in size, this conjugate might represent a small protein of ~ 16 kDa that is linked to a single NEDD8 moiety or, alternatively, it could correspond to a NEDD8 trimer. Other conjugates were also induced in treated cells, in addition to the ~ 25 kDa band, but at lower levels. These NEDD8 conjugates formed rapidly after cells were subjected to oxidative stress (< 5 min), and further increased in abundance following the prolonged treatment of cells with $H_2O_2$ (for up to 1 h; Fig 3B).

When we compared the neddylation pattern generated in wild-type cells after oxidative stress to the one in unstressed NEDP1 knockout cells, we found the resulting neddylation patterns to be strikingly similar; in addition, the neddylation patterns in NEDP1 KO cells did not change following their treatment with $H_2O_2$ (Fig 3B). This suggested that the conjugates generated in wild-type cells after oxidative stress are the same conjugates as those present in NEDP1 knockout cells and are thus likely to represent NEDD8 chains.

To test whether NEDD8 chains serve a role in the oxidative stress response, we subjected NEDP1 KO cells to increasing amounts of hydrogen peroxide and found that deletion of NEDP1 indeed conferred significant resistance to high doses of $H_2O_2$, as measured by CellTiter-Glo assay (Fig 3C). Furthermore, this phenotype was rescued by the re-expression of wild-type NEDP1, proving that this resistance to oxidative stress is due to NEDP1 deletion and is not caused by an off-target effect (Fig 3D).

The known NEDD8 target cullin-3 (Cul3), together with its substrate receptor Kelch-like ECH-associated protein 1 (Keap1), is involved in the cellular response to oxidative stress. Oxidative stress inhibits the Cul3/Keap1-dependent degradation of the transcription factor nuclear factor erythroid 2-related factor 2 (Nrf2), which drives gene expression that protects against cell death (Cullinan *et al*, 2004; Zhang *et al*, 2004). Previous reports have noted some reduction in the activity of cullin-RING ligases in NEDP1 knockout cells (Chan *et al*, 2008; Coleman *et al*, 2017). We also detected slightly reduced cullin neddylation in our NEDP1 knockout cells, including that of Cul3 (Fig 3E and F). This cullin neddylation defect is likely due to the lack of free NEDD8 that is available to conjugate to cullins in the knockout cells. We could not, however, rescue this defect by overexpressing mature NEDD8, as this did not restore free NEDD8 levels. Instead, more non-cullin conjugates were formed, demonstrating that these conjugates form efficiently and preferentially over cullin neddylation (Fig EV3A and B).

Concomitant with lower Cul3 neddylation levels in NEDP1 knockout cells, we detected a small apparent increase of Nrf2 levels by Western blot, which, however, was not statistically significant (Fig EV3C and D). This slight increase might still drive Nrf2-dependent transcription, resulting in resistance to oxidative stress. We therefore performed qPCR experiments to directly measure the RNA levels of Nrf2 and its target NQO1 to test whether the survival phenotype of NEDP1 KO cells is due to the induction of Nrf2-dependent transcription even in the absence of oxidative stress. However, the levels of Nrf2 and NQO1 mRNA were identical to those in wild-type cells (Fig 3G) in contrast to when Nrf2 is stabilized by siRNA-mediated knockdown of Keap1 (Fig EV3E and F). Furthermore, deletion of another cullin regulator, CAND1, which functions by promoting the formation of cullin complexes (Pierce *et al*, 2013; Wu *et al*, 2013; Zemla *et al*, 2013), also leads to a similar stabilization of Nrf2 (Fig EV3C and D), but unlike NEDP1 knockout cells, these cells are not resistant to oxidative stress (Fig 3H). This suggests that any slight stabilization of Nrf2 that was mediated by NEDP1 deletion was not responsible for the resistance to oxidative stress. Instead, we hypothesized that the NEDD8 chains have an active protective function in the cellular response to oxidative stress that is independent of any impairment in CRL activity.

As PARP-1 is required to induce cell death after oxidative stress, we next tested whether PARP-1 activation is impaired in the absence of NEDP1. NEDP1 knockout cells indeed display a strong reduction in high molecular mass PAR polymer formation following oxidative stress, which can be rescued by the re-expression of NEDP1 (Fig 4A and B); this suggests PARP-1 activation is impaired but not completely inhibited (Fig 4A). Furthermore, when WT cells were treated with the small-molecule PARP-1 inhibitors, olaparib or DPQ, cells showed a survival phenotype in response to high doses of $H_2O_2$ that was equivalent to that produced by NEDP1 KO (Figs 4C and EV4A). In addition, the resistance of NEDP1 knockout cells to $H_2O_2$ was not further increased by PARP-1 inhibition (Figs 4C and EV4A). Using immunofluorescence, we also confirmed that the release of AIF from mitochondria and its translocation into the nucleus, a hallmark of PARP-1-dependent cell death, no longer occurs in NEDP1 knockout cells after $H_2O_2$ treatment (Fig 4D and E). This effect is not due to a general reduction in protein levels, as the cellular levels of PARP-1 and of the death inducer AIF were the same in both the U2OS WT and NEDP1 KO cells (Fig 4F). The observed reduction in PARP-1 activity also did not appear to be the result of the neddylation of PARP-1 itself, as it was unmodified in NEDP1 KO cells (Fig 4F), or

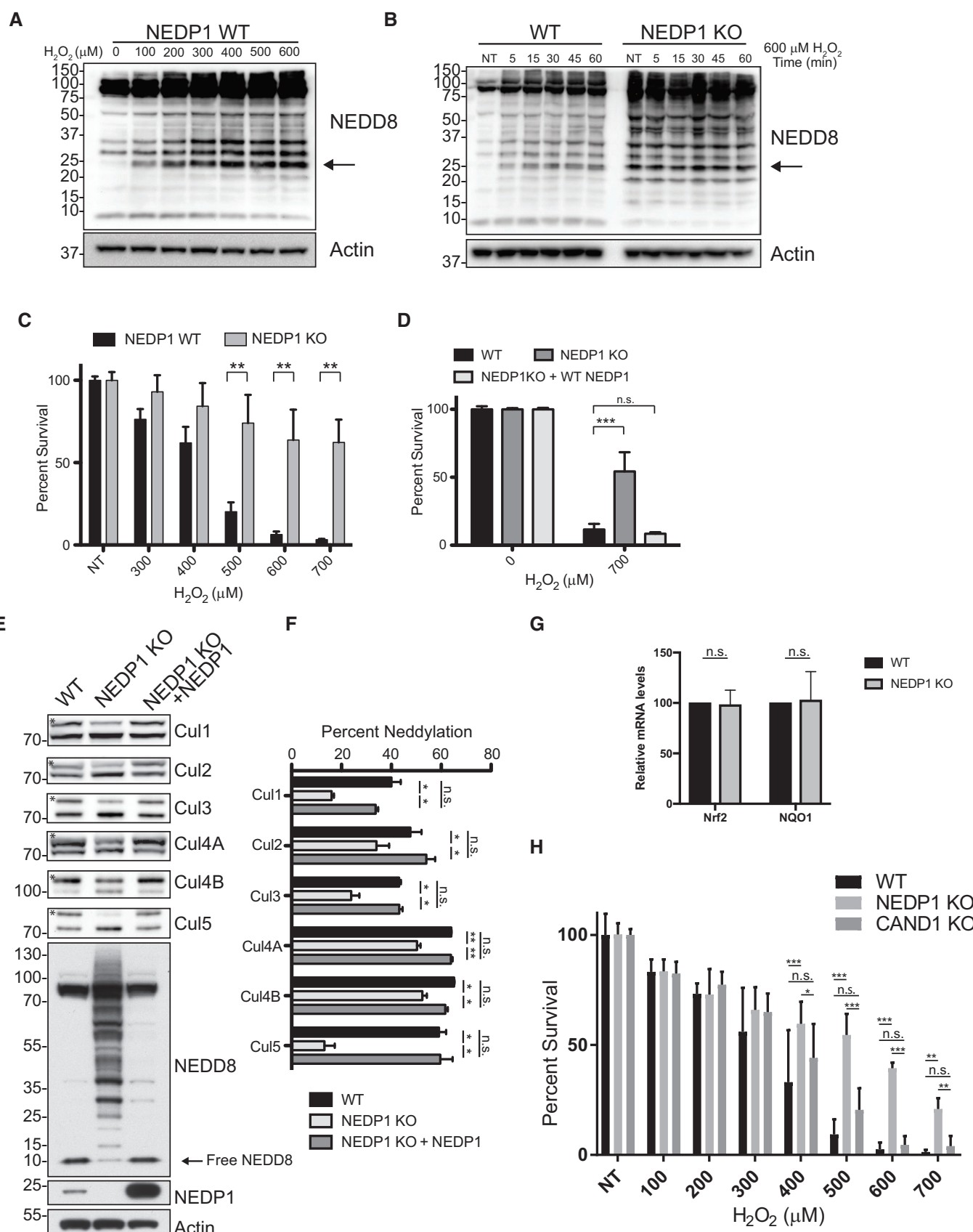

**Figure 3.**

**Figure 3.  NEDD8 chains form in response to oxidative stress and protect NEDP1 KO from H₂O₂.**

A   Non-cullin neddylated species accumulate after oxidative stress in WT U2OS cells with the most highly induced species detected at 25 kDa (as indicated by the arrow). WT U2OS cells were treated with the indicated concentration of $H_2O_2$ for 30 min and then lysed in LDS sample loading buffer and processed for Western blot analysis.

B   (Left) The 25-kDa neddylated species induced by oxidative stress increases with time in WT U2OS cells. U2OS WT or NEDP1 KO cells were treated with 600 μM $H_2O_2$ for the indicated time and then lysed in LDS sample loading buffer and processed for Western blot analysis. (Right) The same 25 kDa NEDD8 species, which is strongly induced in WT cells, is present in untreated NEDP1 KO cells and does not further increase after oxidative stress induced by $H_2O_2$ (as indicated by the arrow).

C   U2OS NEDP1 KO cells are resistant to the PARP-1 inducer, $H_2O_2$. WT and NEDP1 U2OS cells were plated in 96-well plates and after 24 h were treated with the indicated concentration of $H_2O_2$. They were assessed for viability 24 h later by the CellTiter-Glo assay. Graphs represent the mean ± SEM of the percent survival compared to untreated cells. Two-way ANOVA with Bonferroni *post hoc* test: $n = 3$, \*\*$P < 0.0021$.

D   NEDP1 KO cells are re-sensitized to $H_2O_2$, following re-expression of NEDP1 by transient transfection. U2OS cells were plated in 96-well plates and reversed transfected with NEDP1 or an empty vector. Forty hours after plating, cells were challenged with the indicated amount of $H_2O_2$ and cell viability was assessed 24 h later by the CellTiter-Glo assay. Graphs represent the mean ± SEM of the percent survival compared to untreated cells. Two-way ANOVA with Bonferroni *post hoc* test: $n = 3$, \*\*\*$P < 0.001$

E   Neddylation of cullins is decreased in NEDP1 KO cells and is rescued by re-expression of NEDP1 by transient transfection. Western blot analysis from HEK 293 WT, NEDP1 KO cells and NEDP1 KO cells rescued with transient transfection of NEDP1 reveals that upon NEDP1 re-expression, the intensity of the NEDD8 reactive bands is reduced, and the levels of free NEDD8 are increased. NEDP1 re-expression also increased the neddylation of each Cullin. The neddylated band of each Cullin is denoted with a star.

F   Quantification and graph of the mean ± SEM of the percentage neddylation of each Cullin in (E). One-way ANOVA with Bonferroni *post hoc* test: $n = 3$, \*$P < 0.05$, \*\*$P < 0.01$, n.s. denotes not statistically significant.

G   Deletion of *Nedp1* does not lead to induction Nrf2 response genes. RNA was harvested from WT and NEDP1 KO U2OS cells, reversed transcribed to cDNA, and analysed by qPCR for Nrf2 and NQO1 expression. Graph represents the mean ± SEM. One sample *t*-test: $n = 4$, n.s. denotes not statistically significant.

H   Deletion of *Nedp1* but not of *Cand1* in HEK 293 cells results in resistance to cell death from $H_2O_2$. WT, NEDP1 KO and CAND1 KO cells were plated in 96-well plates and treated with the indicated amount of $H_2O_2$. Twenty-four hours after treatment, cell viability was measured using the CellTiter-Glo assay. Graphs represent the mean ± SEM of the percent survival compared to untreated cells. Two-way ANOVA with Bonferroni *post hoc* test: $n = 3$, \*$P < 0.033$, \*\*$P < 0.0021$, \*\*\*$P < 0.001$, n.s. denotes not statistically significant.

when we expressed and immunoprecipitated FLAG-PARP-1 from WT or from NEDP1 KO cells (Fig EV4B). Since Poly(ADP-ribose) is degraded in cells by poly(ADP-ribose) glycohydrolase (PARG), we treated cells with the PARG inhibitor, PDD 17273, to exclude the possibility that the lack of PAR polymer formation is due to the hyperactivation of PARG. While levels of PAR polymer formation were significantly increased in PDD 17273-treated, NEDP1 KO cells, these levels were lower than in WT cells, consistent with an overall reduction, but not with an entire absence of PARP-1 activation in NEDP1 KO cells (Fig EV4C).

We therefore conclude that NEDP1 knockout cells can no longer properly execute PARP-1-dependent cell death due to a significant reduction in PARP-1 activity. This effect is specific to PARP-1-dependent cell death, a caspase-independent cell death pathway (Yu *et al*, 2002; Wang *et al*, 2004), as NEDP1 knockout cells were not resistant to the combined treatment with cycloheximide (10 μg/ml) and TNF-α (100 ng/ml; Fig 5A), a known inducer of apoptosis through caspase activation (Wang *et al*, 2008). In contrast, NEDP1 KO cells were more sensitive than were wild-type cells to treatment with camptothecin (CPT), a drug that stabilizes topoisomerase I (Topo I) linkages to DNA, resulting in DNA double-strand breaks (Fig 5B; Patel *et al*, 2012). The increased sensitivity of NEDP1 KO cells to CPT is consistent with the inhibition of PARP-1 itself, as the inhibition of PARP-1 with the small-molecule inhibitor veliparib has been shown to sensitize cells to Topo I poisons (Patel *et al*, 2012).

Indirect immunofluorescence reveals that the location of NEDD8 is, like PARP-1, mainly nuclear in both WT and KO cells, consistent with NEDD8 being able to directly regulate PARP-1 under physiological conditions (Fig EV4D). We next performed immunoprecipitations to determine whether PARP-1 could interact with NEDD8. Endogenous PARP-1 indeed specifically co-precipitated a single neddylated species that migrated at ~ 25 kDa from NEDP1 knockout cells (Fig 5C), which by size corresponds to the major band induced

after oxidative stress. Similarly, transfection of NEDP1 KO cells with GFP or GFP-PARP-1 followed by anti-GFP immunoprecipitation verified the 25 kDa NEDD8 species specifically bound to GFP-PARP-1 (Fig 5D). To confirm that this is the same species that is generated in cells by oxidative stress, we treated wild-type cells with $H_2O_2$, immunoprecipitated PARP-1 and found that the 25-kDa NEDD8 reactive band again specifically co-precipitated with PARP-1 (Fig 5E), which demonstrates that this interaction occurs under endogenous conditions after stress. We further confirmed this interaction by overexpressing and immunoprecipitating GFP-PARP-1 after oxidative stress. NEDD8 only co-precipitated with GFP-PARP-1 in wild-type cells after treatment with $H_2O_2$ (Fig 5F), while the same species co-precipitated from NEDP1 knockout cells to the same degree irrespective of the presence of oxidative stress (Fig 5F). Interestingly, overexpression of GFP-PARP-1 led to a further significant stabilization of the 25 kDa NEDD8 species following $H_2O_2$ treatment in WT cells, but not in NEDP1 KO cells, which suggests that PARP-1 binding protects the NEDD8 species from NEDP1-dependent cleavage (Fig 5F).

To better understand the interaction between NEDD8 and PARP-1, we subsequently used truncation mutations to map the PARP-1 domain that interacts with NEDD8 and found that NEDD8 bound to the DNA-binding domain of PARP-1 (AA 1–336), but not to its auto-modification and catalytic domain (AA 336–1,014) (Fig 5G). The DNA-binding domain contains three zinc finger domains, and further domain mapping revealed that only the second zinc finger domain (Zn2) interacted with NEDD8 (Fig 5H), which is the zinc finger with the highest affinity for DNA (Langelier *et al*, 2011a). Thus, NEDD8 might inhibit PARP-1 activation by limiting the interaction of PARP-1 with DNA. This suggests that NEDD8 and DNA may compete for the interaction with Zn2 of PARP-1, and indeed, digestion of DNA in NEDP1 KO lysates with micrococcal nuclease (MNase) treatment leads to increased binding of NEDD8 to recombinant GFP-Zn2 (Fig EV4E).

   

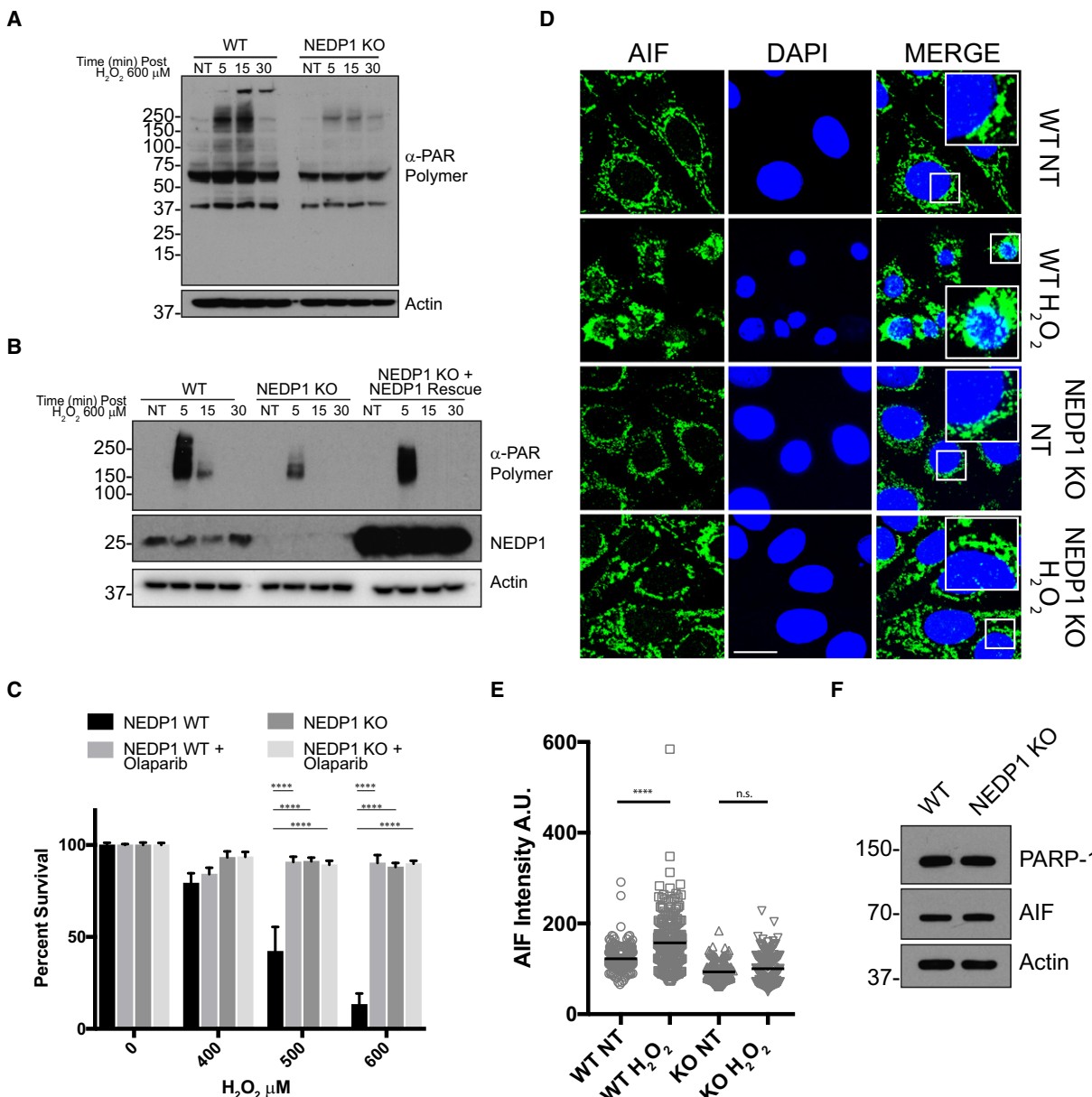

**Figure 4.  NEDP1 knockout inhibits PARP-1 hyperactivation.**

A   PARP-1 activity is reduced in NEDP1 KO cells. Western blot analysis of whole-cell lysates from WT and NEDP1 KO U2OS cells after treatment with 600 μM $H_2O_2$ for the indicated amount of time. PAR polymer generation is induced in WT cells after $H_2O_2$ treatment, but induction is reduced in NEDP1 KO cells.

B   PARP-1 activity is rescued by re-expression of NEDP1 in NEDP1 KO cells. Western blot analysis of whole-cell lysates from WT and NEDP1 KO U2OS cells transfected with the indicated constructs. Seventy-two hours after transfection, cells were treated with the indicated amount of $H_2O_2$ and processed for Western blot analysis. PAR polymer generation is reduced in NEDP1 KO cells but can be rescued by the transient re-expression of NEDP1.

C   PARP-1 inhibitor olaparib protects WT cells from $H_2O_2$ treatment to the same extent as NEDP1 KO. U2OS cells were plated in 96-well plates and 24 h later pre-treated with olaparib (10 μM) for 1 h before treatment with the indicated concentration of $H_2O_2$. Twenty-four hours later, cell viability was measured using the CellTiter-Glo assay. NEDP1 deletion protects cells from $H_2O_2$ treatment to the same extent as PARP-1 inhibition in WT cells. Olaparib does not provide additional protection to NEDP1 cells from $H_2O_2$ treatment. Graphs represent the mean ± SEM of the percent survival compared to untreated cells. Two-way ANOVA with Bonferroni *post hoc* test: $n = 3$, ****$P < 0.0001$.

D   AIF translocation to the nucleus is impaired in NEDP1 KO cells. WT and NEDP1 KO cells were plated on coverslips and 24 h later were treated with $H_2O_2$ (600 μM) for the indicated amount of time and then processed for immunofluorescence analysis using α-AIF antibodies and DAPI staining. PARP-1-dependent cell death is induced in WT U2OS cells, as indicated by translocation of AIF from the mitochondria to the nucleus (scale bar 10 μm).

E   Total nuclear AIF intensity from (D) was measured with ImageJ and plotted as a vertical scatter plot with the group mean intensity indicated with a black bar. AIF has translocated to the nucleus after WT cells were exposed to $H_2O_2$ but not in NEDP1 KO cells. Kruskal–Wallis test with Dunn's multiple comparison *post hoc* test: $n = 3$, ****$P < 0.0001$, n.s. denotes not statistically significant.

F   NEDP1 KO cells express normal levels of PARP-1 and AIF. Western blot analysis of whole-cell lysates from WT and NEDP1 KO U2OS cells shows that the proteins necessary for the induction of PARP-1-dependent cell death are present at equal levels in both cell lines.

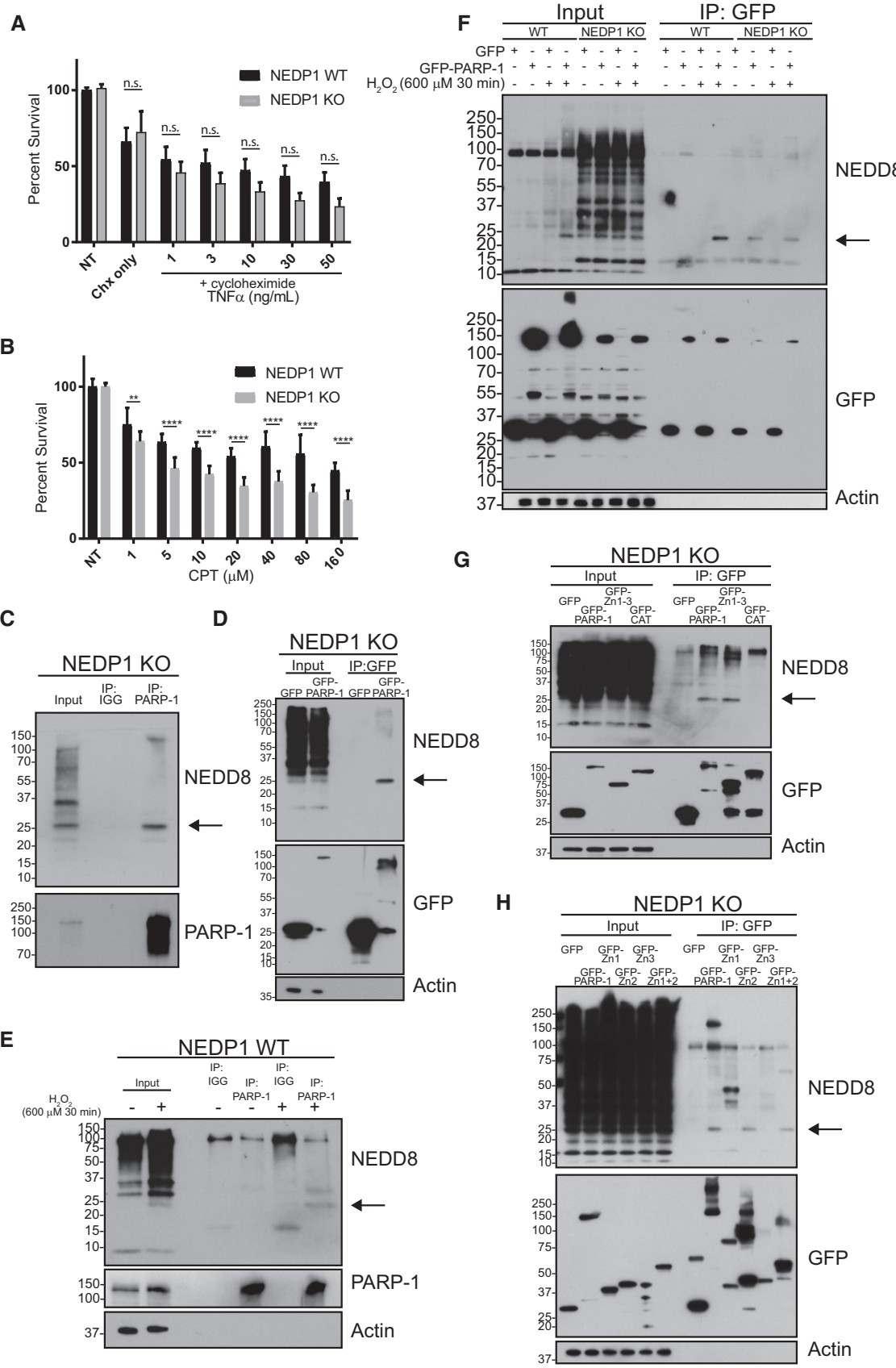

**Figure 5.**

**Figure 5.  NEDD8 trimers bind to PARP-1 to prevent activation of parthanatos.**

A   NEDP1 KO U2OS cells are not resistant to induction of apoptosis from combined treatment with TNF-α and cycloheximide. U2OS cells were plated in 96-well plates and incubated with cycloheximide (10 μg/ml) before addition of the indicated amount of TNF-α. Cell survival was determined 24 h later using the CellTiter-Glo assay. Graphs represent the mean ± SEM of the percent survival compared to untreated cells. Two-way ANOVA with Bonferroni *post hoc* test: n = 3, n.s. denotes not statistically significant.

B   NEDP1 KO U2OS cells are not resistant to treatment with camptothecin (CPT). WT U2OS and NEDP1 KO cells were plated in 96-well plates and treated with the indicated amount of CPT. Forty-eight hours after exposure cell survival was measured using the CellTiter-Glo assay. Graphs represent the mean ± SEM of the percent survival compared to untreated cells. Two-way ANOVA with Bonferroni *post hoc* test: n = 3, **P < 0.0021, ****P < 0.0001.

C   NEDD8 trimers co-immunoprecipitate with endogenous PARP-1 in NEDP1 KO cells. Lysates were prepared from U2OS NEDP1 KO cells, and immunoprecipitation was performed with PARP-1-Trap or GFP-Trap as a negative control. Bound proteins were resolved by SDS–PAGE and processed for Western blot analysis with the indicated antibodies.

D   NEDD8 trimers bind to GFP-PARP-1. U2OS NEDP1 KO cells were transfected with GFP or with GFP-PARP-1, and 24 h later, cell lysates were collected. Immunoprecipitation was then performed with GFP-Trap, and bound proteins were resolved by SDS–PAGE and processed for Western blot analysis with the indicated antibodies. GFP-PARP-1 but not GFP alone can co-immunoprecipitate NEDD8 trimers from NEDP1 KO cells.

E   NEDD8 trimers form in WT U2OS cells after treatment with H$_2$O$_2$ and co-immunoprecipitate with endogenous PARP-1 but not with immunoglobulin control (IGG). WT U2OS cells were left untreated or treated with H$_2$O$_2$ (600 μM) for 30 min and were harvested for immunoprecipitation and Western blot analysis as in (C).

F   NEDD8 trimers bind to GFP-PARP-1. WT or U2OS NEDP1 KO cells were transfected with GFP or with full-length GFP-PARP-1 and 24 h later treated as indicated with H$_2$O$_2$ (600 μM) for 30 min before cell lysates were collected. Immunoprecipitation was then performed with GFP-Trap, and bound proteins were resolved by SDS–PAGE and processed for Western blot analysis with the indicated antibodies. GFP-PARP-1 but not GFP alone can co-immunoprecipitate NEDD8 trimers from WT cells only after H$_2$O$_2$ treatment or from both treated or untreated NEDP1 KO cells.

G   NEDD8 trimers bind to the DNA-binding domain of PARP-1 (Zn1–3) but not to its automodification and catalytic domain (CAT). NEDP1 KO U2OS cells were transfected with GFP, GFP-PARP-1, GFP fused to the DNA-binding domain of PARP-1 (AA 1–336) or with GFP fused to the PARP-1 automodification and catalytic domain (AA 336–1014). Twenty-four hours post-transfection, cell lysates were collected and immunoprecipitation with GFP-Trap was performed. Bound proteins were eluted and resolved by SDS–PAGE analysis, followed by Western blot analysis with the indicated antibodies.

H   NEDD8 trimers specifically bind to the second zinc finger of PARP-1. NEDP1 KO U2OS cells were transfected with GFP, GFP-PARP-1, or with GFP fused to the Zn1 domain (AA 1–96), the Zn2 domain (AA 97–215), the Zn3 domain (AA 216–336) or the Zn1 + 2 domains (AA 1–215) of PARP-1. Twenty-four hours post-transfection, cell lysates were collected and immunoprecipitation with GFP-Trap was performed. Bound proteins were eluted and resolved by SDS–PAGE analysis, followed by Western blot analysis with the indicated antibodies.

In addition to its roles in DNA repair, PARP-1 also inhibits the transcription of some NF-κB-responsive genes, including CXCL10, independently of its catalytic activity, but via its ability to directly bind to DNA (Carrillo *et al*, 2004). Consistent with the NEDD8 trimers blocking PARP-1-DNA interaction, we found that CXCL10 transcription was de-repressed in NEDP1 knockout cells after TNF-α treatment (Fig EV5A), which could be rescued by re-expressing NEDP1 (Fig EV5B).

We conclude from these results that NEDD8 binds to the Zn2 domain of PARP-1 to attenuate its activation either through directly competing with PARP-1 binding to DNA or by impairing the assembly of the independent domains of PARP-1 required for full activation. To confirm that the NEDD8 species binding to PARP-1 is indeed a trimeric unanchored chain, we used a GST fusion of the PARP-1 zinc fingers 1 and 2 (GST-Zn1 + 2) (Fig EV5C) to purify trimeric NEDD8 from NEDP1 knockout cells and analysed the bound proteins by 2D gel electrophoresis. 2D gel electrophoresis has previously been used to characterize free ubiquitin chains (Amerik *et al*, 1997). As for ubiquitin, the isoelectric point of an unanchored NEDD8 chain would be the same as that of mono-NEDD8 (Halligan, 2009) and would only shift higher on the molecular mass dimension of a 2D gel. 2D gel electrophoresis of WT cell lysate confirmed that monomeric NEDD8 migrated to its predicted isoelectric point (Fig 6A). We found that a proportion of the ~ 25 kDa NEDD8 purified via the GST-PARP-1 Zn1 + Zn2 focused to the same isoelectric point as mono-NEDD8 (Fig 6B), demonstrating that it is indeed an unanchored NEDD8 chain, which specifically interacts with the PARP-1 Zn fingers (Fig EV5C and D). However, the chain was not homogenous, and, in fact, most NEDD8 species focused to multiple isoelectric points, indicating that NEDD8 chains have a more negative charge than would be expected of a NEDD8 trimer (Fig 6B). The fact that the isoelectric point shifted to the left, indicated that

the chain is post-translationally modified to either introduce new negative charges or to mask positive charges. Furthermore, as the non-modified trimer seemed to be the least abundant, this experiment also suggested that such a modification might be required for optimal binding of NEDD8 to PARP-1. We thus re-analysed our mass spectrometry data for post-translational modifications of NEDD8 and found significant amounts of lysine acetylation on K11, K22, K33 and K48. Lysine acetylation would indeed shift the protein to a lower isoelectric point, as it would mask the positive charges on NEDD8.

From these results, we hypothesized that the acetylation of NEDD8 may serve to mask the positively charged lysine residues of NEDD8 to facilitate its binding to the PARP-1 Zn2 domain. To confirm whether changes in acetylation influence the interaction of NEDD8 with PARP-1, we overexpressed the de-acetylases HDAC1 or HDAC2 in NEDP1 KO cells, which indeed led to a decrease in the amount of NEDD8 trimer bound to exogenously expressed and precipitated GFP-PARP-1 Zn1 + 2 (Fig 6C). Likewise, treatment of NEDP1 KO cells with the histone de-acetylase inhibitor sodium butyrate (NaB) increased the amount of NEDD8 bound to GFP-PARP-1 Zn1 + 2 (Fig 6D). Similarly, when we used bacterially expressed GFP-Zn2 to purify tri-NEDD8 from NEDP1 knockout lysate, we detected an increase in the amount of bound tri-NEDD8 after prior treatment of the cells with NaB (Fig 6E). Furthermore, the overexpression of either HDAC1 or HDAC2 not only reduced the binding of tri-NEDD8 to PARP-1, but also rescued PARP-1 activation in NEDP1 knockout cells, as measured by PAR polymer generation after oxidative stress (Fig 6F), providing strong evidence that the direct binding of tri-NEDD8 to PARP-1 Zn2 inhibits PARP-1 activation and that acetylation facilitates the interaction of PARP-1 and NEDD8, possibly by through direct acetylation of NEDD8 lysines.

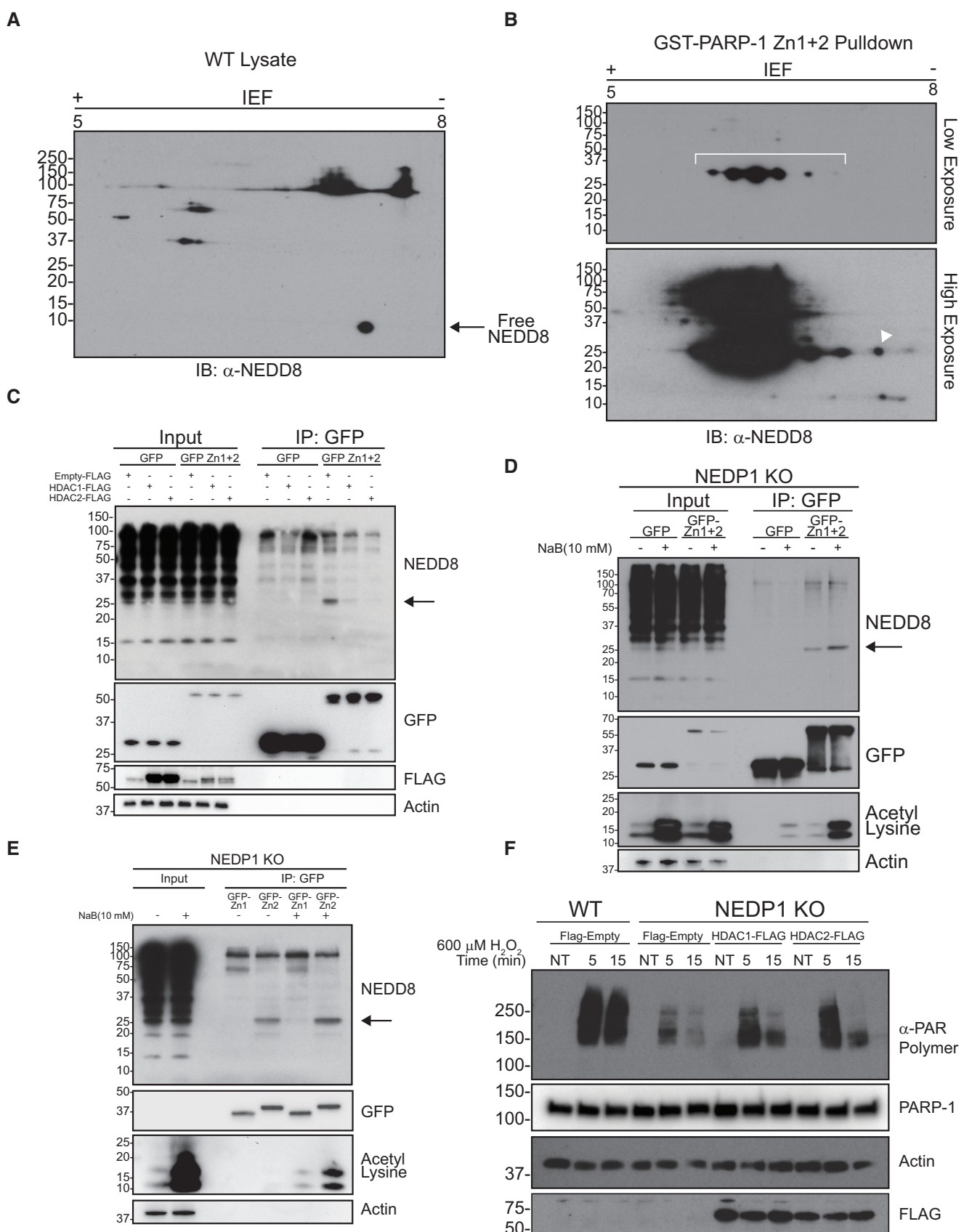

**Figure 6.**

◄

**Figure 6. NEDD8 trimers are acetylated to increase binding to the second zinc finger of PARP-1.**

A   2D gel electrophoresis of lysate from WT cells followed by Western blot analysis with α-NEDD8 antibody indicates free NEDD8 migrates near its predicted isoelectric point (pI) of 6.59.

B   2D gel electrophoresis of pulldowns from NEDP1 KO lysate with GST-Zn1 + 2 followed by Western blot analysis with α-NEDD8 antibody. Some of the 25-kDa NEDD8 band migrates to the same isoelectric point as unconjugated NEDD8 (denoted by white triangle), indicating the species is most likely an unanchored NEDD8 trimer. However, the majority of NEDD8 trimer migrates to multiple spots of lower pI (indicated in the brackets), which suggests that NEDD8 trimers undergo a post-translational modification that either adds a negative charge or blocks a positive charge on the NEDD8 trimer.

C   Overexpression of HDAC1 or HDAC2 decreases PARP-1-NEDD8 trimer binding. NEDP1 KO U2OS cells were transfected with GFP or GFP fused to the first two zinc finger domains of PARP-1 (GFP-Zn1 + 2). Cells were also co-transfected, as indicated, with empty FLAG vector, HDAC1-FLAG, or HDAC2-FLAG. Twenty-four hours post-transfection, cells were harvested and immunoprecipitation was performed on the lysates with GFP-Trap. Immunoprecipitated proteins were resolved by SDS–PAGE followed by Western blot analysis with the indicated antibodies.

D   HDAC inhibition increases PARP-1-NEDD8 trimer binding. NEDP1 KO U2OS cells were transfected with GFP or GFP fused to the first two zinc finger domains of PARP-1 (GFP-Zn1 + 2). Twenty hours later, cells were left untreated or treated with the HDAC inhibitor sodium butyrate (NaB) (10 mM) for 4 h. Cells were harvested, and immunoprecipitation was performed on the lysates with GFP-Trap. Immunoprecipitated proteins were resolved by SDS–PAGE followed by Western blot analysis with the indicated antibodies.

E   HDAC inhibition increases PARP-1-NEDD8 trimer binding. NEDP1 KO U2OS cells were left untreated or treated with the HDAC inhibitor sodium butyrate (NaB) (10 mM) for 4 h. Cells were harvested, and lysates were incubated with recombinant Zn1-GFP or Zn2-GFP (25 nM) for 1 h followed by immunoprecipitation with GFP-Trap. Immunoprecipitated proteins were resolved by SDS–PAGE followed by Western blot analysis with the indicated antibodies.

F   Overexpression of HDAC1 or HDAC2 in NEDP1 KO cells can rescue the induction of PAR polymer following $H_2O_2$ treatment. U2OS WT or NEDP1 KO cells were transfected with the indicated FLAG vectors. Forty-eight hours post-transfection, cells were treated with $H_2O_2$ (600 μM) for the indicated amount of time and harvested directly in sample loading buffer. Lysates were resolved by SDS–PAGE and processed for Western blot analysis with the indicated antibodies.

One question that remains to be addressed is how oxidative stress is sensed and how it leads to the formation of poly-NEDD8. We reasoned that NEDP1 itself might be directly inhibited in the presence of reactive oxygen species, which could occur by direct oxidation of the catalytic cysteine of NEDP1. Indeed, the disassembly of *in vitro* NEDD8 chains by NEDP1 was strongly inhibited by the presence of even low amounts of $H_2O_2$ (Fig 7A). This was in stark contrast to the forward reaction, mediated by NAE/UBE2M. NAE/UBE2M also contain active-site cysteines, but their activity was largely unaffected by the presence of even high amounts of $H_2O_2$ (Fig 7B). Thus, in a cellular context, NEDD8 chains may be continuously formed by NAE/UBE2M, but rapidly cleaved by NEDP1. After oxidative stress, however, NEDP1 catalytic activity could be directly impaired by active-site cysteine oxidation, while the forward reaction would remain unaffected, allowing for the accumulation of NEDD8 chains and the rapid attenuation of PARP-1 hyper-activation.

## Discussion

NEDD8 is a major regulator of cellular function through its role in activating cullin-RING ligases by mono-neddylation. Many non-cullin roles for NEDD8 have been proposed over the years, but they have remained controversial as a result of the realization that NEDD8 can enter the ubiquitin pathway when overexpressed. We now show that, when expressed at endogenous levels, NEDD8 can efficiently form non-cullin conjugates through the bona fide NEDD8 pathway. Our data suggest that most of these conjugates are in the form of poly-NEDD8 chains, which are, at least to some degree, unanchored. In the absence of NEDP1, these chains form extremely efficiently at the expense of cullin neddylation, suggesting that UBE2M generates them preferentially, depleting the pool of free, unconjugated NEDD8. This, we believe, may largely explain the defects in cullin neddylation that we and others have observed in NEDP1 knockout cells, since there may simply not be enough NEDD8 available to also fully neddylate cullins.

Evolutionarily, UBE2M has retained the ability to form poly-NEDD8 chains efficiently, suggesting that these chains perform an important function. Also, NEDD8 contains 8/9 lysine residues that can be used to form chains and which are conserved from *Caenorhabditis elegans* to humans (Fig EV5E). If the mono-neddylation of cullins was the only important function of NEDD8, one might expect that more of these lysines would have been lost without the evolutionary pressure to maintain them or that the ability of UBE2M to form chains would have been lost. Our data now suggest that NEDD8 chain formation is an important part of the cell's response to oxidative stress and that these chains function to counteract the hyper-activation of PARP-1. While we identify a role of the trimeric unanchored NEDD8 species on binding to and inhibiting PARP-1, it may also serve other functions. Furthermore, additional NEDD8 conjugates also form and it is tempting to speculate that these conjugates serve other roles and bind to different targets. If so, rapid poly-NEDD8 formation could coordinate numerous cellular responses by serving as a platform for oxidative stress signalling. The attenuation of PARP-1 activation might thus only be one outcome of poly-NEDD8 stabilization.

Our data show that NEDP1 enzymatic activity is much more sensitive to reactive oxygen species than is the forward reaction that is mediated by NAE and UBE2M. It is thus extremely tempting to speculate that NEDP1 is directly inhibited by reactive oxygen species to allow poly-NEDD8 to form as a rapid response to oxidative stress. This would not be unlike the Keap1 system that also rapidly responds to oxidative stress by the direct oxidation of cysteines within Keap1 (Cullinan *et al*, 2004; Zhang *et al*, 2004). This oxidation inactivates Keap1, leading to the stabilization of the transcription factor Nrf2, which under non-stressed conditions is normally degraded via Keap1. Nrf2 then drives a protective transcriptional program (Ishii *et al*, 2000). Future work will need to substantiate these assumptions, but it is feasible that poly-NEDD8 inhibits PARP-1 to delay the commitment to PARP-1-dependent cell death until other protective mechanisms, such as the transcriptional program mediated by Nrf2 stabilization, can be launched.

Tri-NEDD8 very specifically binds to the Zn2 domain of PARP-1, which is a critical domain for PARP-1 activation. PARP-1 can

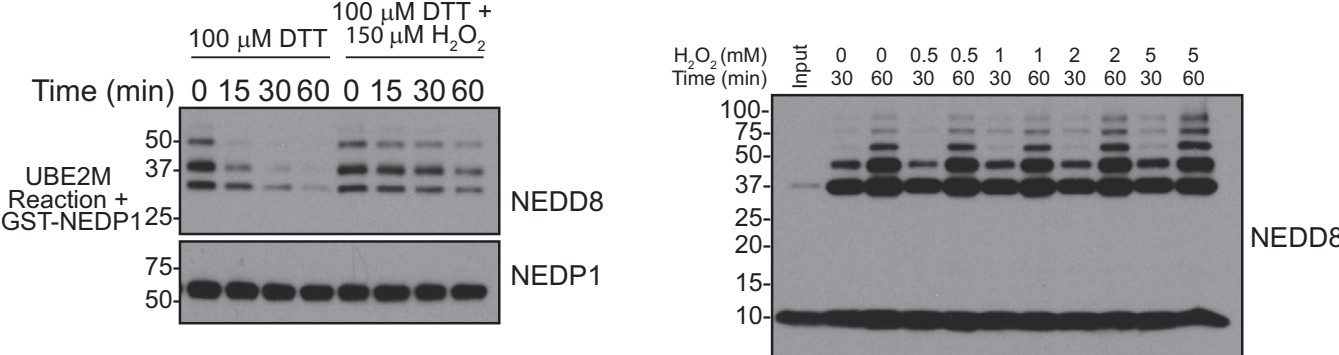

**Figure 7. NEDP1 activity but not UBE2M activity is inhibited by oxidative stress.**

A  *In vitro* de-neddylation by NEDP1 is sensitive to oxidation. Poly-NEDD8 reactions were prepared with NAE, UBE2M, and NEDD8 as in Fig 1E. NEDD8 reactions were stopped by the addition of NAE inhibitor MLN4924 (3 μM). NEDD8 reactions were incubated with GST-NEDP1 at 30°C for the indicated amount of time under reducing (50 μM DTT) or oxidizing conditions (50 μM DTT + 150 μM $H_2O_2$). Reactions were stopped by the addition of LDS sample loading buffer and resolved by SDS–PAGE before processing for Western blot analysis with the indicated antibodies.

B  *In vitro* neddylation reactions with UBE2M are insensitive to $H_2O_2$. Poly-NEDD8 reactions were prepared with NAE, UBE2M and NEDD8 as in Fig 1E. Before reactions were started, increasing concentrations of $H_2O_2$ (as indicated) were added to the NEDD8 reactions. Subsequently, the reactions were incubated at 30°C for the indicated amount of time. Reactions were stopped by the addition of LDS sample loading buffer and resolved by SDS–PAGE before processing for Western blot analysis with the indicated antibodies.

recognize multiple types of DNA damage including the single-strand breaks (SSB) that occur following oxidation of DNA. Zn2 binds to DNA with the highest affinity (Langelier *et al*, 2011a) and is the first domain to interact with the SSB on the 3′ side of the DNA lesion (Eustermann *et al*, 2015). Subsequently, Zn1 binds the 5′ side of the SSB, which induces an allosteric assembly of the separate domains of PARP-1 into a catalytically active enzyme (Eustermann *et al*, 2015). PARP-1 automodifies itself with negatively charged PAR polymer to drive its release from DNA (Satoh & Lindahl, 1992; Murai *et al*, 2012). Once free from DNA, PARP-1 interacts with PARG to degrade the automodifications to allow for PARP-1 binding to a new DNA lesion (Ueda *et al*, 1972; Miwa *et al*, 1974). NEDD8 trimers bind to the second zinc finger domain of PARP-1, which likely inhibits the critical first step of PARP-1 activation. When sufficient DNA damage occurs, like after $H_2O_2$ treatment, the NEDD8 chains may ensure that the time PARP-1 is unbound from DNA is long enough for PARG to completely degrade PAR automodifications on PARP-1. If modified PARP-1 rebinds too quickly to DNA damage, then existing PAR automodifications may otherwise be extended to produce the toxic form of PAR.

We identified an unexpected complexity when performing 2D gel electrophoresis of the NEDD8 trimers that bind to PARP-1. The majority of NEDD8 trimers were post-translationally modified to produce net negative shifts in the isoelectric point of the trimer, and our mass spectrometry data identified NEDD8 acetylation as its only abundant modification. The acetylation of positively charged lysines would explain this decrease in the isoelectric point of trimeric NEDD8 and also the multiple shifts in its isoelectric point that we observe. The fact that most of NEDD8 that binds to the PARP-1 DNA-binding domain is acetylated indicates that the masking of

NEDD8's positive charges might be necessary for it to interact with this domain, particularly given that this domain normally binds to negatively charged DNA. This is substantiated by the observation that the global modification of acetylation levels, through treatment with sodium butyrate or via HDAC1/2 overexpression, influenced the binding of NEDD8 to PARP-1. HDAC inhibition has been shown to increase DNA damage and to sensitize cancer cells to chemotherapeutics (Eot-Houllier *et al*, 2009). However, increased cellular acetylation via HDAC inhibition has also been reported to be protective in various mouse models of neurodegeneration (Didonna & Opal, 2015) and to protect neurons from oxidative stress within 2 h (Langley *et al*, 2008). This effect was largely attributed to the increased transcription of protective genes as a result of histone acetylation. The acetylation-dependent inhibition of PARP-1 hyperactivation via NEDD8 trimers provides another potential mechanism by which HDAC inhibitors could provide cellular protection from oxidative stress.

The finding that trimeric unanchored NEDD8 species appear to bind to and to inhibit PARP-1 is a significant one. While PARP-1 inhibition is an attractive therapeutic target in cancer owing to its role in regulating DNA damage, it also prevents cell death in certain conditions. In fact, *Parp-1* knockout mice are resistant to ischaemic stroke, as well as to LPS-induced septic shock, and to streptozotocin-induced diabetes (Eliasson *et al*, 1997; Burkart *et al*, 1999; Kuhnle *et al*, 1999; Masutani *et al*, 1999; Oliver *et al*, 1999; Pieper *et al*, 1999). Furthermore, the genetic deletion or pharmacological inhibition of PARP-1 in human embryonic stem cell-derived neurons was found to be protective in a model of oxygen glucose deprivation (Xu *et al*, 2016). Thus, PARP-1 is an attractive drug target to prevent cell death and our findings suggest that NEDD8 regulates this aspect of PARP-1.

# Materials and Methods

### Antibodies

Rb α-NEDP1 1:1,000 (Thermo PA5-31033), Sh α-NEDP1 1:1,000 (DSTT S378D against AA1–212), Ms α-DCUN1D1 1:1,000 (Sigma Clone 3D7), Sh α-DCUN1D2 1:1,000 (DSTT S995C against AA1–60) incubated with recombinant DCNL1 (AA 1–45) peptide to block cross reaction, Sh α-DCUN1D3 1:2,000 (DSTT S996C against AA1–60), Sh α-DCUN1D4 1:2,000 (DSTT S997C against AA1–119), Sh α-DCUN1D5 1:2,000 (DSTT S998C against AA1–124), Rb α-NEDD8 1:1,000 (Abcam ab81264), Sh α-UBE2M 1:1,000 [DSTT S432D against full length (mouse)], Sh α-UBE2F 1:1,000 (DSTT S438D against AA1–70), Ms α-Actin 1:1,000 (Millipore MAB1501, Ms α-Ubiquitin 1:1,000 (Millipore MAB1510), Rb α-RBX1 1:200 (Thermo PA5-16282), Rb α-CAND1 1:1,000 (CST 7433S), Rb α-CSN5 1:1,000 (Abcam ab12323), Rb α-CSN8 1:1,000 (Abcam EPR5139), Rb α-UBA3 1:1,000 (Epitomics 5157-1), Ms α-AppBp1/ULA1 1:1,000 (BD 611865), Rb α-Cul1 1:1,000 (Life Tech 718700), Rb α-Cul2 1:5,000 (Life Tech 700179), Sh α-Cul3 1:1,000 (DSTT S067D, AA554–768), Sh α-Cul3 1:1,000 (DSTT S464D AA544–768), Sh α-Cul4A 1:3,500 (DSTT S087D AA1–124), Rb α-Cul4A 1:1,000 (CST 2699S), Sh α-Cul4B 1:1,000 (DSTT S070D AA1–162), Sh α-Cul5 1:1,000 (DSTT S073D AA577–689), Ms α-IκBα 1:1,000 (CST 4814), Ms α-pIκBα 1:1,000 (CST 9246 clone 5A5), Ms α-PAR Polymer 1:1,000 (Enzo Lifescience ALX-804-220-R100 clone 10H), Rb α-PARP-1 1:1,000 (CST 9532), Rb α-AIF 1:1,000 (Abcam ab 32516), Rb α-NRF2 1:1,000 (Abcam ab62352), Rb α-HIF-1α 1:1,000 (Novus Biologicals NB 100-654), Ms α-FLAG M2 1:2,000 (Sigma F3165), Ms α-GFP 1:5,000 (Abcam ab184519); Rb α-PARG 1:1,000 (CST 66564S); Rb α-Keap1 1:1,000 (CST 8047).

### Chemicals

TNF-α, PeproTech (300-01A); HALO-link resin, Promega (G1913); MLN7243, Active Biochemicals (A-1384); MLN4924, Active Biochemicals (A-1139); $H_2O_2$, Sigma (216763); Camptothecin, Sigma (C9911); Cycloheximide, Sigma (C1988); DPQ, Calbiochem (300270); Cell-Titer-Glo, Promega (G7570); Olaparib, Cambridge Biosciences (CAY10621); PARG inhibitor PDD 17273, Tocris (5952); Sodium Butyrate, Sigma (303410); GFP-Trap_A, Chromotek (gta), PARP-1-Trap_A, Chromotek (xta); phosSTOP, Sigma (4906845001); cOmplete EDTA-free protease inhibitor cocktail (Roche-11836170001); Pepstatin A, Sigma (P5318), Bestatin hydrochloride, Sigma (B8385); Deoxyribonuclease I, Sigma (D5025); LDS Sample Buffer, Life Technologies (NP0007); Colloidal Coomassie, Expedeon (ISB1L); IPTG, Formedium (IPTG025); Micrococcal Nuclease, NEB (M0247S).

### Plasmids

his6-HALO NEDP1(C163A) DSTT DU28042
his6-HALO NEDP1(C163A D29W A98K G99K) DSTT DU28008
pCMV5-NEDP1 DSTT DU28199
GST-NEDP1 DSTT DU23262
Frt/TO N-FLAG empty DSTT DU13236
Frt/TO N-GFP empty DSTT DU45825
Frt/TO C-GFP empty DSTT DU41574
Frt/TO C-FLAG empty DSTT DU43547
GST empty DSTT DU49206

gRNAs for NEDP1 and CAND1 were generated by mutagenesis PCR of pEsgRNA (Munoz *et al*, 2014) using gRNA sequence CCCCGTAGTCTTGAGTTACA, which targets the beginning of the NEDP1 protein coding sequence in exon 5, or of gRNA sequence TCACCTAAAGTCCTTGTCGC, which targets the first exon of CAND1.

NEDP1 gRNA F 5′-GGAAAGGACGAAACACCGCCCCGTAGTCTTGA
GTTACAGTTTTAGAGCTAGAAAT-3′
NEDP1 gRNA R 5′-ATTTCTAGCTCTAAAACTGTAACTCAAGACTAC
GGGGCGGTGTTTCGTCCTTTCC-3′
CAND1 gRNA F 5′-GGAAAGGACGAAACACCTCACCTAAAGTCCTT
GTCGCGTTTTAGAGCTAGAAAT-3′
CAND1 gRNA R 5′-ATTTCTAGCTCTAAAACGCGACAAGGACTTTA
GGTGAGGTGTTTCGTCCTTTCC-3′

Full-length PARP-1 (NM_001618.3) was amplified from HUVEC cDNA and cloned into the N-terminal GFP-tagged vector. For domain mapping, individual zinc finger domains were cloned into the C-terminal GFP vector and the catalytic domain was cloned into the N-terminal GFP vector. GST-PARP-1 ZN1 + 2 was cloned in pGEX-6P-2 (GE Lifesciences). NSUN2 (NM_017755.5) was amplified from HUVEC cDNA and cloned into the N-terminal FLAG empty vector. HDAC1 (NM_004964.2) and HDAC2 (NM_001527.3) were both amplified from HUVEC cDNA and cloned into the C-terminal FLAG empty vector.

PARP-1 Full(NM_001618.3) F 5′-ATGGCGGAGTCTTCGGATAAGC
TCT-3′
PARP-1 Full(NM_001618.3) R 5′-TAGTACGCGGCCGCTTACCACA
GGGAGGTCTTAAAATTGAATTTCAGTTTCAGC-3′
PARP-1 Zn1 (AA1-96) F 5′-ATGGCGGAGTCTTCGGATAAGCTCT-3′
PARP-1 Zn1 (AA1-96) R 5′-GTACTAGCGGCCGCGCCTGTCACTCCT
CCAGCTTCCG-3′
PARP-1 Zn2 (AA97-215) F 5′-TAGTACGGATCCAAAGGCCAGGAT
GGAATTGGTAGC-3′
PARP-1 Zn2 (AA97-215) R stop 5′-GTACTAGCGGCCGCCTATCCAT
CCACCTCATCGCCTTTTCT-3′
PARP-1 Zn2 (AA97-215) R no stop 5′-GTACTAGCGGCCGCTCCATC
CACCTCATCGCCTTTTCT-3′
PARP-1 Zn3 (AA216-336) F 5′-TAGTACGGATCCGTGGATGAAGTG
GCGAAGAAGAAATCTA-3′
PARP-1 Zn3 (AA216-336) R 5′-GTACTAGCGGCCGCTGGGGTTACC-
CACTCCTTCC-3′
PARP-1 C-terminal (AA336-1014) F 5′-TAGTACCCCGGGAAGGAAT
TCCGAGAAATCTCTTACCTCAA-3′
PARP-1 C-terminal (AA336-1014) R 5′-TAGTACGCGGCCGCTTACC
ACAGGGAGGTCTTAAAATTGAATTTCAGTTTCAGC-3′
NSUN2(NM_017755.5) F 5′-AATAATAGAATTCATGGGGCGGCGGT
CGCGGG-3′
NSUN2(NM_017755.5) R 5′-TATTATTGCGGCCGCTCACCGGGGT
GGATGGACCCCC-3′
HDAC1(NM_004964.2) F 5′-AAAGGATCCATGGCGCAGACGCAGG
GCACCCGGAGGAAAGT-3′
HDAC1(NM_004964.2) R 5′-TACGCGGCCGCGGCCAACTTGACCTC
CTCCT-3′
HDAC2(NM_001527.3) F 5′-TACGGATCCATGGCGTACAGTCAAGG
AGGCGGCAAAAAA-3′

HDAC2(NM_001527.3) R 5′-TACGCGGCCGCGGGGTTGCTGAGCTG
TTCTGATTTGGTTC-3′

## RNA isolation and qPCR

HEK 293 cells were stimulated with TNF-α (PeproTech, 10 ng/ml) for 4 h at 37°C and harvested with RNeasy (Qiagen, Venlo, Netherlands) following the manufacturer's protocol. Following RNA purification, RNA concentration was measured with a Nano-Drop (Thermo Fisher Scientific) and 1 μg of RNA was used to generate cDNA using the High-Capacity cDNA Reverse Transcription Kit (Invitrogen #4368814). qPCRs were performed with Brilliant III Ultra-Fast SYBR Green qPCR master mix (Agilent 600882) on a Bio-Rad CFX96. The $2^{-\Delta\Delta Ct}$ method (Livak & Schmittgen, 2001) was used for RNA quantification using 18S ribosomal RNA or GAPDH mRNA as the control RNA standard. Primers used were as follows:

18S F 5′-GTAACCCGTTGAACCCCATT-3′
18S R 5′-CCATCCAATCGGTAGTAGCG-3′
CXCL10 F 5′-GCTGATGCAGGTACAGCGT-3′
CXCL10 R 5′-CACCATGAATCAAACTGCGA-3′
IκBα F 5′-GATCCGCCAGGTGAAGGG-3′
IκBα R 5′-GCAATTTCTGGCTGGTTGG-3′
Nrf2 F 5′-AGACGGTATGCAACAGGACA-3′
Nrf2 R 5′-AGTTTGGCTTCTGGACTTGGA-3′
NQO1 F 5′-TCACCGAGAGCCTAGTTCCG-3′
NQO1 R 5′-TGGCATAGTTGAAGGACGTCC-3′
GAPDH F 5′-GAAATCCCATCACCATCTTCCAGG-3′
GAPDH R 5′-GTACCTCTTCCGACCCCGAG-3′

## Proteins

The purification of ubiquitin, NEDD8, UBE2M, UBE2F, and of the APPBP1/UBA3 heterodimer was performed as previously described in Kelsall *et al* (2013). pET28a 6His-HALO NEDP1 was transformed into BL21 cells and purified with $Ni^{2+}$-Sepharose (GE Healthcare). GST-NEDP1 and GST vectors were transformed into BL21 cells and purified with Glutathione Sepharose 4B (GE Healthcare). $His_6$-USP1/UAF1 was obtained from the Division of Signal Transduction Therapy (DSTT) (Dundee, UK) product, DU23056. The plasmids encoding GST-PARP-1 Zn1 + 2, GST-PARP-1 Zn1-GFP and GST-PARP-1 Zn2-GFP were transformed into BL21 Rosetta 2 (DE3)-competent cells (Novagen 71400). Cells were grown in 2YT media supplemented with 100 μM $ZnSO_4$, and protein expression was induced with 0.2 mM IPTG at 16°C for 20 h. Cells were pelleted by centrifugation, and pellets were resuspended in lysis buffer (25 mM HEPES pH 8.0, 500 mM NaCl, 0.1% NP-40, 1 μM pepstatin A, 1 μM bestatin, and cOmplete EDTA-free protease inhibitor cocktail). Pellets were sonicated; then, 5 mM $MgCl_2$ and DNase I were added to 200 Kunitz/ml; and the cell suspension was incubated with rotation at 4°C for 1 h. Cell lysate was spun at $20,000 \times g$ for 30 min, and supernatant was collected. GST fusion proteins were then purified with Glutathione Sepharose 4B (GE Healthcare). The GST tag was cleaved from GST-PARP-1 Zn1-GFP and GST-PARP-1 Zn2-GFP with PreScission Protease (GE Healthcare). Bound DNA was removed from Zn1-GFP and Zn2-GFP via a Heparin column (GE Healthcare) as previously described (Langelier *et al*, 2011b).

## Cell culture

HEK 293 and U2OS cells were obtained from ATCC and were grown in DMEM High Glucose (Gibco, Life Technologies) supplemented with 10% FBS and 3 mM L-Glutamine (Gibco, Life Technologies) in a 37°C incubator with 5% $CO_2$. Cell lines were routinely checked for mycoplasma contamination. siRNA-mediated knockdown of DCNL1-5 was carried out as previously described (Keuss *et al*, 2016). The same protocol was followed for siRNA-mediated knockdown of UBE2M (Dharmacon Smart Pool L-004348-00), of UBE2F (Dharmacon Smart Pool L-009081-01), and of Keap1 (Dharmacon Smart Pool L-012453-00). HEK 293 and U2OS Cas9 knockouts were generated in Trex Flp-In Flag-Cas9 cell lines as in Keuss *et al* (2016) and Munoz *et al* (2014).

Whole-cell lysates were prepared with buffer A: 50 mM Tris–HCl pH 7.5, 150 mM NaCl, 5% glycerol, 0.5% NP-40, 0.5% sodium deoxycholate, 0.1% SDS, 1 mM EDTA, 3 mM 1,10-phenanthroline, and 15 mM iodoacetamide (IAA), supplemented with phosStop (Roche) and cOmplete mini protease inhibitor (Sigma). Lysates were sonicated and centrifuged at 4°C for 20 min at $17,000 \times g$, and supernatants were collected. Protein concentration was measured using the BCA assay using the manufacturer's protocol (Pierce, Thermo Scientific). To immunoprecipitate the FLAG-tagged proteins, cells were harvested as above in buffer A. Supernatants were incubated with anti-FLAG-M2 magnetic beads (Sigma) for 3 h at 4°C. Beads were then washed three times with 0.5 ml of buffer A without inhibitors and then resuspended in reaction buffer (50 mM Tris–HCl pH 7.5, 150 mM NaCl, 5% glycerol, and 1 mM DTT). Beads were split evenly between three tubes, and either GST-NEDP1, his6-USP1/UAF1 or no protein was added to each tube. Reactions were then incubated at 30°C for 30 min before being stopped with LDS sample loading buffer. To immunoprecipitate endogenous PARP-1 and GFP fusion proteins, cells were harvested with buffer B: 50 mM Tris–HCl pH 7.5, 150 mM NaCl, 5% glycerol, 0.5% NP-40, 5 mM $MgCl_2$, 0.2 mM $CaCl_2$, 1 μM pepstatin A, 1 μM bestatin supplemented with phosStop and cOmplete mini protease inhibitor. Lysates were sonicated for six pulses of 0.5 s. Subsequently, DNase I (200 Kunitz/ml) was added to lysates and incubated for 1 h at 4°C with rotation. Lysates were centrifuged at 4°C for 20 min at $17,000 \times g$, and supernatants were collected. GFP-Trap beads or PARP-1-Trap beads were incubated with lysates for 3 h at 4°C with rotation. Next, agarose beads were washed three times with 0.5 ml of wash buffer (50 mM Tris–HCl pH 7.5, 150 mM NaCl, 5% glycerol and 0.5% NP-40). Bound proteins were eluted with LDS sample loading buffer and heated at 95°C for 5 min with 10 mM DTT. For GFP pulldown with HDAC overexpression, cells were lysed with buffer B with 2.7 mM KCL. For GFP pulldown after MNase treatment, cells were lysed with buffer B with 75 mM NaCl, 5 mM $CaCl_2$, 15 mM IAA, 3 mM OPT and 4,000 units of MNase (NEB) per ml of lysate and incubated at 25°C for 30 min.

For the analysis of Cullin neddylation, cells were lysed directly in urea loading buffer (25 mM Bis-Tris pH 6.8, 8 M urea, 6% SDS, 5% glycerol) and DNA was fragmented by passage through a homogenizer column (Omega Bio-tek). Proteins were subsequently reduced with 10 mM DTT at 37°C for 10 min before loading onto SDS–PAGE gels. For the analysis of PAR polymer formation after $H_2O_2$, cells were lysed directly in sample loading buffer (25 mM Tris pH 8.5, 6% SDS, 5% glycerol, 10 μM olaparib and 5 μM PARG

inhibitor PDD 17273) and DNA was fragmented by passage through a homogenizer column (Omega Bio-tek). Proteins were subsequently reduced with 10 mM DTT at 95°C for 5 min before loading onto SDS–PAGE gels.

For cell viability analysis of U2OS cell lines, cells were plated at 15,000 cells per well of a 96-well plate and 24 h later were treated with the indicated concentration of $H_2O_2$ continuously for 24 h. Where indicated, cells were treated with DMSO, olaparib, or DPQ for 1 h before treatment with $H_2O_2$, and continuously thereafter. After 24 h, cell viability was determined using the CellTiter-Glo assay (Promega). To assess the viability of U2OS cell lines after treatment with camptothecin, cells were plated at 5,000 cells per well of a 96-well plate and 24 h later were treated with the indicated concentration of camptothecin at 37°C. After 48 h, cell viability was measured using the CellTiter-Glo Assay. To analyse cell death after TNFα + cycloheximide treatment, U2OS cells were plated at 10,000 cells per well of a 96-well plate. Twenty-four hours after plating, cells were treated with cycloheximide (10 μg/ml) for 1 h and then treated with the indicated amount of TNF-α. Twenty-four hours after treatment, cell viability was measured using the CellTiter-Glo Assay (Promega).

## Western blotting

Cell lysates were resolved on Bis-Tris gels (homemade or NuPAGE 4–12%) with MOPS buffer (Formedium MOPS-SDS5000) and transferred to nitrocellulose or 0.2-μm PVDF membranes. Membranes were blocked for 1 h at RT with 5% milk, or with 5% BSA in TBST, and then incubated with the indicated antibodies overnight at 4°C. Membranes were then washed 3 × 10 min with TBST and incubated with HRP-conjugated secondary antibodies for 1 h at RT and then washed 3 × 10 min with TBST. Membranes were incubated with chemiluminescent substrate (EMD Millipore WBKLS0500) and exposed to Konica Blue X-ray film or, for quantification, were imaged using a ChemiDoc Imaging system (Bio-Rad). Images were quantified with ImageJ (NIH).

## Statistics

All statistical analysis was performed with GraphPad Prism using the statistical tests indicated in the figure legends.

## Immunofluorescence

U2OS cells were plated on coverslips coated with 10 μg/ml Poly-D-lysine (P6407-5 mg Sigma) and 24 h later were left untreated or treated with 600 μM $H_2O_2$ for 9 h at 37°C. Cells were then washed once with ice-cold phosphate-buffered saline (PBS) and then fixed with ice-cold methanol for 1 min. Coverslips were washed three times with PBS and then blocked with 3% BSA in PBS for 1 h at room temperature. Next, coverslips were incubated with primary antibodies (Rb α-AIF, Abcam ab32516, 1:1,000 or Rb α-NEDD8, Abcam ab81264, 1:1,000) diluted in PBS with 3% BSA for 1 h at room temperature. Coverslips were washed three times with PBS and then incubated with Alexa 488 conjugate anti-rabbit secondary antibody (1:1,000, Thermo A-11034) for 1 h at room temperature. DAPI (300 nM, Life Technologies, D1306) staining was performed for 5 min, followed by one wash with PBS. Coverslips were

mounted onto slides with ProLong Gold Antifade reagent (Life Technologies) and imaged on a Zeiss LSM 710 confocal microscope. Quantification of nuclear AIF intensity was performed using ImageJ v1.50g (National Institute of Health).

## *In vitro* NEDD8 chain formation

NEDD8 chains were made *in vitro* by incubating NAE (0.15 μM), UBE2M/2F (10 μM) and NEDD8 (20 μM) in reaction buffer (50 mM Tris–HCl pH 8.0, 200 mM NaCl, 20% glycerol, 10 mM $MgCl_2$, 10 mM ATP, 0.6 mM DTT) for 3 h at 30°C. Reactions were stopped by the addition of LDS sample loading buffer.

## Mass spectrometry

For HALO pulldown for mass spectrometry analysis, NEDP1 KO HEK 293 cells were grown on 150-$cm^2$ plates, scraped in ice-cold PBS, collected and centrifuged at $500 × g$ for 5 min. Cell pellets were resuspended in buffer A without SDS, and lysates were spun at $17,000 × g$ for 20 min. The supernatant was collected and incubated with HALO-Link resin (Promega) conjugated to HALO-NEDP1 C163A or DAGC mutant for 3 h at 4°C. HALO resin was washed three times with buffer A and three times with 50 mM Tris–HCl without salt or detergent. Bound proteins were eluted with 100 μl of a 50:50 mixture of acetonitrile and 0.1% formic acid for a total of five elutions. Eluate was collected and dried in a speed vac (Thermo Scientific). Protein pellets were resuspended in LDS sample loading buffer and resolved by SDS–PAGE on a 4–12% Bis-Tris gel (Novex, Life Technologies). Gels were stained with InstantBlue colloidal Coomassie (Expedeon). Bands were then excised, washed with water, dehydrated in acetonitrile and rehydrated in 50 mM Tris–HCl pH 8.0. Gel slices were alkylated with 20 mM chloroacetamide, dehydrated in acetonitrile and then transferred to 50 mM triethylammonium bicarbonate. Gel bands were incubated with trypsin (5 μg/ml) overnight at 30°C. Peptides were extracted with acetonitrile, gel bands were incubated in 0.1% trifluoroacetic acid (TFA), and then, peptides were extracted two more times with acetonitrile. Extracted peptides were dried in a speed vac and then resuspended in 0.1% TFA/water. Samples were analysed on a LTQ Orbitrap Velos Pro mass spectrometer (Thermo Fisher Scientific) coupled to an Ultimate 3000 UHPLC system with 15-cm Acclaim PepMap 100 analytical column (75 μm ID, 3 μm C18 from Thermo Scientific) with an addition Pepmap trapping column (100 μm × 2 cm, 5 μm C18; Thermo Fisher Scientific). Acquisition settings were lockmass of 445.120024, MS1 with 60,000 resolution, top 20 CID MS/MS using rapid scan, monoisotopic precursor selection, unassigned charge states and $z = 1$ rejected, dynamic exclusion of 60s with repeat count of 1. One-hour linear gradients were performed from 5% solvent B to 35% solvent B (solvent A: 0.1% formic acid, solvent B: 80% acetonitrile 0.08% formic acid). Raw files were processed in Proteome Discoverer 2.0 (Thermo Scientific), with Mascot 2.4.1 (Matrix Science),and subsequently processed in Scaffold 4.4.6 (Proteome Software) Searches were performed with a peptide tolerance of 10 ppm (monoisotopic) and a fragment tolerance of 0.60 Da (monoisotopic) or with MaxQuant v1.5.7.4 for label-free quantitative and iBAQ analysis. Settings were fixed modifications of carbamidomethyl (C), variable modifications of oxidation (M), dioxidation (M), LRGG (K) and GlyGly (K). Protein

identifications were filtered with a 1% FDR. Raw files were re-searched separately with variable modification for acetylated peptides acetyl (K) or phosphorylated peptides phospho (STY); both searches included the fixed modifications of carbamidomethyl (C), variable modifications of oxidation (M) and dioxidation (M).

For analysis of *in vitro* neddylation reactions, the reactions were resolved on a 4–12% Bis-Tris gel (Novex, Life Technologies), stained with InstantBlue colloidal Coomassie (Expedeon), and processed for in-gel trypsin digest, as described above.

### 2D gel electrophoresis

To prepare lysate from wild-type U2OS cells, one T-150 flask was harvested with 5 ml trypsin-EDTA (Life Technologies). Detached cells were collected with fresh media and centrifuged at $500 \times g$ for 5 min. Cell pellets were washed two times with 10 ml of PBS and then lysed with 8 M urea and 0.5% CHAPS. Protein concentration was determined by BCA assay using the manufacturer's protocol (Peirce, Thermo Scientific). To remove salt and impurities, protein was precipitated with acetone. Four times the sample volume of −20°C acetone was added and the solution was incubated at −20 for 2 h. The sample was then centrifuged at 4°C for 20 min at $17,000 \times g$. The supernatant was decanted, and the pellet was washed with four volumes of 80% acetone, followed by centrifugation at 4°C for 5 min at $17,000 \times g$. The supernatant was decanted, the pellet was briefly centrifuged again, and residual acetone was removed. The pellet was dried at room temperature for 5 min. Pellets were resuspended with rehydration buffer (6 M urea, 2 M thiourea, 4% CHAPS, 0.5% IPG buffer (Bio-Rad). 65 mM DTT and trace bromophenol blue) to 30 μg/ml. Subsequently, 90 μg of protein was added to a 7 cm, pH 5–8 IPG strip (Bio-Rad) and allowed to rehydrate on the strip for 14 h, followed by focusing for 12 h for a total of 22 kVhrs. For two-dimensional electrophoresis, separation strips were equilibrated with equilibration buffer (Bio-Rad) with 50 mM DTT for 15 min. The buffer was replaced and the strips equilibrated again for 15 additional minutes. IPG strips were loaded onto precast 4-20% TGX gels (Bio-Rad) and processed for Western blot analysis.

For 2D gel electrophoresis analysis of GST pulldowns, USOS NEDP1 KO cells were harvested with lysis buffer B, as described above for the endogenous PARP-1 immunoprecipitation. Next, 20 mg of lysate was used for each pulldown with 100 μg GST or 100 μg GST-PARP-1 Zn1 + 2 preconjugated to Sepharose 4B beads. The beads and lysates were incubated at 4°C for 2 h with rotation. The beads were then washed four times with wash buffer (50 mM Tris–HCl pH 7.5, 150 mM NaCl, 0.5% NP-40, 5% glycerol), followed by three washes with salt-free buffer (50 mM Tris–HCl pH 7.5). Bound proteins were eluted with 8 M urea and 0.5% CHAPS and were processed for isoelectric focusing, as described above.

## Data availability

The mass spectrometry data from this publication have been deposited to the PRIDE Archive (www.ebi.ac.uk/pride/archive/) with the identifier PXD011928.

**Expanded View** for this article is available online.

## Acknowledgements

We thank David G Campbell and Joby Varghese (MRC-PPU, Dundee) for help with mass spectrometry sample preparation and data analysis. Mathias Saur for assistance in creating gRNA vectors for NEDP1 and CAND1. We would like to thank the Division of Signal Transduction Therapy (DSTT) (Dundee) for technical support, and, specifically, Nicola Wood for DNA cloning, James Hastie and Hilary McLauchlan for antibody production and Axel Knebel for protein purification. MT was funded by the Medical Research Council (MC_UU_12016/5). Work in the TK laboratory is supported by grants from Motor Neurone Disease Scotland and the British Council BIRAX initiative.

## Author contributions

MJK performed most experiments. RH designed NEDP1 constructs, performed HALO-NEDP1 pulldowns and sample preparation for the HEK 293 mass spectrometry experiment and performed some cell viability experiments. OH performed microscopy of AIF translocation. RG processed mass spectrometry samples and performed database searches. RB performed 2D gel electrophoresis. MJK, RG, RH, MT and TK analysed mass spectrometry results. MJK, RH and TK designed experiments. MJK and TK wrote the manuscript with input from all authors. RH and TK conceived the project, and TK supervised the work.

## Conflict of interest

The authors declare that they have no conflict of interest.

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
