## [Review Process File · The EMBO Journal]

Unanchored tri-NEDD8 inhibits PARP-1 to protect from oxidative stress-induced cell death

Matthew J. Keuss, Roland Hjerpe, Oliver Hsia, Robert Gourlay, Richard Burchmore, Matthias Trost, Thimo Kurz.

Review timeline:

Submission date:	11 th June 2018
Editorial correspondence:	29 th July 2018
Additional correspondence from author:	30 th July 2018
Editorial Decision:	13 th November 2018
Revision received:	3 rd December 2018
Editorial Decision:	21 st December 2018
Revision received:	10 th January 2019
Accepted:	28 th January 2019

Editor: Hartmut Vodermaier

Transaction Report:

Editorial correspondence

29th July 2018

Thanks for your patience during the arbitrating review of your transferred manuscript. I have now received input from two trusted expert advisors of our journal, who had been given access to your manuscript as well as your response letters to the original reviewers' reports. As you will see from the comments copied below, these two arbitrators are rather divided in their opinion. Arbitrator 1 considers the demonstration of endogenous NEDD8 conjugation beyond cullin ligases important and most previously raised concerns well-answered, and would not insist on decisive testing of the conjecture that direct (stoichiometric) PARP-1 binding by NEDD8 trimers competes with DNA binding and activation of PARP-1. On the other hand, arbitrator 2 finds such conclusions insufficiently supported on multiple levels and overall raises very similar major concerns as previous referee 3. A particularly important issue is the cropping of gels and weak interaction data in Figure 5, which would have to be decisively clarified prior to any further consideration. In addition, this arbitrator also criticizes several other pieces of evidence for the generation and direct involvement of unanchored chains in PARP-1 regulation.

Before taking further decisions, I would at this stage like to give you an opportunity to answer to the comments of our arbitrators, both with a tentative point-by-point response on what could be done to answer the points of both of them, and definitely by providing the full source data for the blots criticized by original referee 3 and by our arbitrator. Based on this, we will then decide on whether we could pursue publication in The EMBO Journal further or not.

REFeree REPORTS.

Arbitrator #1 (Report for Author)

I have reviewed the manuscript without prior reading of previous reviewer comments.

This manuscript, in my view, provides the strongest evidence for a function for NEDD8 beyond modification of cullin-based ubiquitin ligases. What are almost certainly free NEDD8 chains based on the mass spectrometry analysis and 2D gel analysis, accumulate in the nedp1delta mutant and to a much lesser extent in WT cells. They show these chains are made by the standard NEDD8 conjugation machinery, not ubiquitin pathway enzymes as had occurred in earlier studies using overexpressed NEDD8. A reasonable case is made here that these NEDD8 chains are induced in WT cells by oxidative stress (hydrogen peroxide) and negatively regulate PARP-1 activation in vivo and PARP-1's ability to induce cell death. H2O2 is likely to directly impair the NEDP1 protease.

The weakest element is probably the conclusion that acetylated NEDD8 trimers directly bind PARP-1, blocking its activation in vivo by preventing PARP-1 binding to (damaged) DNA. However, there is enough evidence favoring this conclusion that I think the manuscript in its present form is appropriate for publication in the EMBO J. I would have liked to see an in vitro deacetylation of NEDD8 after purifying with PARP-1 Zn1-Zn2 followed by a re-examination of NEDD8 species on 2D gels, but this is not essential. Purifying sufficient (acetylated) NEDD8 trimers, doing quantitative binding studies with PARP-1, and showing this association blocks DNA binding will be challenging but are excellent goals for the future.

Some of the previous reviewers wanted the full quantitative in vitro analysis with NEDD8 trimers; I believe this is too steep a demand. Most of the other criticisms were reasonably rebutted by Dr. Kurz.

Arbitrator #2 (Report for Author)

This paper suggests that mechanism by which oxidative stress leads to inhibition of the NEDD8-cleaving enzyme NEDP1, thereby resulting in free NEDD8 chains that could inhibit PARP and PARP-dependent cell death. The paper contains one important set of data, i.e. the demonstration that NEDP1-deleted cells fail to fully activate PARP. However, the mechanistic analysis is superficial and lacks many required essential experiments. Whether effects of NEDP1-deletion or inhibition on PARP are direct through untethered NEDD8 chains or indirect is not clear. There are also issues with data presentation, i.e. the cropping of gels whether the full MW range would need to be shown or the separation of input and IP lanes. In the current state, I would suggest the authors go back and improve their manuscript, before submitting it again.

Major issues:

One clear problem is that they actually do not provide direct evidence that small MW NEDD8-conjugates are untethered chains. The data based on correlation with in vitro reactions (which are much less capable of assembling unlinked chains as compared to modified UBE2M) is very weak. Their cellular assay provides evidence for chain assembly, but lack of finding an acceptor substrate does not indicate that these chains are untethered (it might simply be that their experimental conditions led to cleavage of these chains from targets during lysis, a very common event in PTM pathways).

The enrichment of PARP, while potentially there, is certainly not striking compared to the many other proteins that remain unlabeled.

Fig. 3A: not clear why they find that the 25kD band is the most accumulating band in response to H2O2. Whether these conjugates indeed correspond to free NEDD8-trimers has not been tested.

Fig. 3: the arguments that protection from H2O2-induced death in NEDP1 deleted cells is independent of the established CUL3-KEAP1 axis is indirect and not very convincing. It certainly does not provide strong evidence for a notion that free Nedd8-trimers mediate this effect.

The phenotype of aberrant PAR formation in NEDP1-knockout cells is nice and indicates misregulation of PARP or related enzymes. This is also supported by their epistasis analysis.

Fig. 5C-G really require the whole NEDD8 blot - it is incomplete, and potentially very misleading, to only crop out the 25kD band. Fig.5D is overprocessed, suggesting that this interaction with this particular band is very unstable. As shown, these experiments cannot be interpreted at all, yet they would be an essential point for the authors. Frankly, I do not understand why they cut out the 25kD band, as it does suggest that other NEDD species bind and the mechanism of regulation might not be as straightforward as suggested by the authors. In addition, inputs and IPs have to be on the same gel - the current way of depicting the experiment is misleading and inappropriate.

The argument that Nedd8-binding to PARP inhibits the DNA-binding activity of the latter is very weak. To make such a broad statement, the authors would need to show that a significant fraction of PARP is associating with NEDD8 - which is not supported by the preliminary interaction studies shown in Fig. 5. They would also need to show directly that such competition takes place - binding to an overlapping domain, in the absence of any known Kd values, does not constitute strong evidence.

The decrease in binding to 25kD-NEDD8 (again, full blot needs to be shown) by HDAC-expression and the apparent increase by HDAC inhibition are minor and thus, evidence for a role of acetylation in regulating this interaction is weak. The potential role of acetylation in this pathway has not been addressed.

The last figure does not add much to the paper, and certainly does not provide a molecular mechanism for how oxidative stress in cells might trigger the proposed pathway.

Additional correspondence - author

30th July 2018

Thank you for your email and the comments. My postdoc is not back from his summer break until later next week, so I won't be able to reply in detail until then, but I thought I'd just acknowledge receipt of your email and give you my initial thoughts. I'm happy that referee 1 is so positive about our submission, but also appreciate the concerns of referee 2. I'll be able to give you a point-by-point response to their concerns once I have discussed them with my postdoc. I'd like to mention though that we will be certainly able to provide full length blots and they were cropped for the figures due to space constraints. The previous journal requires the submission of the original source data for all papers anyways, hence why we more liberally cropped them for the actual figures.

As far as Referee 2's other points go, I'll have a talk with my postdoc regarding these before I respond in detail. I'd like to mention already though that I find they're rather "soft" comments without any suggested experiments, but only a rather general disbelief in our results. I would have appreciated if they had actually suggested some experiments that we could do to address their concerns.

Personally, at the moment, I feel that it is a matter of what you think is novel about our paper and how far you'd want us to go to proof every single bit of the hypothesis beyond any doubt. The paper is long and extensive as it is and yes, we could work another two years on it and make it even longer and more data dense. As referee one mentions, this manuscript provides "the strongest evidence for a function of NEDD8 beyond modification of cullin-based ubiquitin ligases" to date. I wholeheartedly agree with that, and as a NEDD8 researcher this is hugely exciting to me. We show there is something else but cullin, we have a phenotype and I think we provide good evidence for how this phenotype can come about. Based on the data, which I think is generally pretty high quality, that's the best interpretation we can come up with. My gut feeling is that the only way we can satisfy referee 2 is with the in vitro reconstitution. It's the only way we'll get the Kd values and can do competition experiments, but as referee 1 also agrees, that is asking a bit much. And if we had all that, I think the paper would actually be a strong contender for Nature or Science. Also, I entirely disagree with the comment that our NEDD8 chains could have "fallen off" a substrate during sample prep. For most of the experiments, safe the IPs, we put SDS buffer directly onto the cells and I have no idea how this would allow for cleavage of NEDD8 chains off a substrate. Also, which enzymes are they thinking would cleave those chains? There is no NEDP1 in those cells and as we know from the recent structures, the signalosome can't do it. The chains are formed within minutes after H2O2 treatment, which also suggests that they don't fall "off" some cryptic substrate.

It's obviously hard to prove absence of anything, but I think this comment is pretty harsh given all the evidence we produce to the contrary. I am happy though to discuss cryptic substrates and as yet un-identified de-neddylases that work in the presence of SDS as an alternative explanation of what we see in the discussion.

If you're concerned that we're "overselling" the PARP angle a bit too much, I'd be happy to tone this down a bit in how we write the paper. We could certainly qualify our discussion a bit more and even change the title if that's what's needed.

1st Editorial Decision

13th November 2018

Thank you again for your discussions on ways to answer the referee reports related to your manuscript on NEDD8 chains and PARP inhibition, and for your updates with additional data and ongoing experiments. In light of this more conclusive support for key aspects of the study, we shall be happy to consider a revised version of the manuscript for publication in The EMBO Journal. I would therefore like to now formally invite you to submit such a revision, modified along the lines proposed in our previous discussions, alongside a comprehensive and diligent point-by-point response to our reviewers' comments.

1st Revision - authors' response

3rd December 2018

Response to reviewers

We appreciate the comments of the reviewers and thank them for pointing out some of the limitations of our earlier submission. We have now revised the manuscript and below provide a point-by-point response to the raised criticism:

Response to Referee #1

This manuscript, in my view, provides the strongest evidence for a function for NEDD8 beyond modification of cullin-based ubiquitin ligases. What are almost certainly free NEDD8 chains based on the mass spectrometry analysis and 2D gel analysis, accumulate in the nedp1delta mutant and to a much lesser extent in WT cells. They show these chains are made by the standard NEDD8 conjugation machinery, not ubiquitin pathway enzymes as had occurred in earlier studies using overexpressed NEDD8. A reasonable case is made here that these NEDD8 chains are induced in WT cells by oxidative stress (hydrogen peroxide) and negatively regulate PARP-1 activation in vivo and PARP-1's ability to induce cell death. H2O2 is likely to directly impair the NEDP1 protease.

We are pleased with this assessment and agree that uncovering a non-cullin role for NEDD8 is exciting.

The weakest element is probably the conclusion that acetylated NEDD8 trimers directly bind PARP-1, blocking its activation in vivo by preventing PARP-1 binding to (damaged) DNA. However, there is enough evidence favoring this conclusion that I think the manuscript in its present form is appropriate for publication in the EMBO J. I would have liked to see an in vitro deacetylation of NEDD8 after purifying with PARP-1 Zn1-Zn2 followed by a re-examination of NEDD8 species on 2D gels, but this is not essential. Purifying sufficient (acetylated) NEDD8 trimers, doing quantitative binding studies with PARP-1, and showing this association blocks DNA binding will be challenging but are excellent goals for the future.

We agree that purifying and analysing acetylated NEDD8 trimers would be the gold standard for supporting our hypothesis. This has, however, proven challenging and was not feasible to achieve in the timescale for these revisions. We have, however, added additional evidence in support for the involvement of acetylation in regulating the interaction between NEDD8 and PARP1: We have bacterially expressed and purified a tagged version of the second zinc finger

of PARP1 and used it to purify tri-NEDD8 from cell extract of NEDP1 knockout cells in pulldown experiments (Fig 6E). Under standard conditions, this efficiently and specifically purifies the tri-NEDD8 chain (Fig 6E). When cells are treated with HDAC inhibitors prior to lysis (resulting in generally increased cellular acetylation), more tri-NEDD8 binds to Zn2 using the above described assay (Fig 6E). While these results don't unequivocally prove that direct acetylation of NEDD8 is required for binding to Zn2, it shows that increased acetylation in general leads to increased binding of NEDD8 to Zn2 and the most straight forward explanation would be that direct acetylation of NEDD8 is required for binding, further supporting our hypothesis.

Response to Referee #2

This paper suggests that mechanism by which oxidative stress leads to inhibition of the NEDD8-cleaving enzyme NEDP1, thereby resulting in free NEDD8 chains that could inhibit PARP and PARP-dependent cell death. The paper contains one important set of data, i.e. the demonstration that NEDP1-deleted cells fail to fully activate PARP. However, the mechanistic analysis is superficial and lacks many required essential experiments. Whether effects of NEDP1-deletion or inhibition on PARP are direct through untethered NEDD8 chains or indirect is not clear. There are also issues with data presentation, i.e. the cropping of gels whether the full MW range would need to be shown or the separation of input and IP lanes. In the current state, I would suggest the authors go back and improve their manuscript, before submitting it again.

We appreciate that the referee considers our demonstration that NEDP1 deleted cells fail to fully activate PARP-1 an important finding. We're equally excited about this discovery. While we disagree with their assessment that our initial analysis was superficial, we have now further improved our data to mitigate the referee's concerns as outlined below in our specific responses. We would like to emphasise already here though that we unequivocally shown that NEDD8 interacts with PARP-1 after oxidative stress in wildtype cells and in unstressed NEDP1 knockout cells. Given the effect of NEDP1 deletion on PARP-1 activation, we find it reasonable to assume that this interaction is important for the effect on PARP-1, especially as we show that when the interaction of NEDD8 with PARP-1 is abrogated (through over-expression of HDACs), the activity of PARP-1 is restored in NEDP1 knockout cells (Fig 6F). It's impossible to prove the absence of another regulatory mechanism, but that is always the case, and we believe that our hypothesis is sufficiently supported by the data we provide. Future work will show if our hypothesis needs to be adjusted, but we do believe that the observation that NEDD8 interacts with PARP-1 after oxidative stress and in NEDP1 knockout cells, and that PARP-1 is inhibited under these conditions, will stand the test of time.

Specific comments:

1.

One clear problem is that they actually do not provide direct evidence that small MW NEDD8-conjugates are untethered chains. The data based on correlation with in vitro reactions (which are much less capable of assembling unlinked chains as compared to modified UBE2M) is very weak. Their cellular assay provides evidence for chain assembly, but lack of finding an acceptor substrate does not indicate that these chains are untethered (it might simply be that their experimental conditions led to cleavage of these chains from targets during lysis, a very common event in PTM pathways).

We don't agree with the referee that our in vitro reactions provide weak evidence for the ability of NEDD8 to form unanchored chains. These reactions are performed with highly purified protein. The smallest protein in the reaction, aside from NEDD8, is the E2 UBE2M. There are clearly conjugates formed at lower molecular masses than UBE2M, which based on the constituents of the reaction, can only be NEDD8~NEDD8 linkages. These conjugates run at the expected molecular masses of free NEDD8 chains and are visible by coomassie staining of the gel, indicating that they are sufficiently abundant. Based on this data we have no doubt that UAE/UBE2M can form free NEDD8 chains – and we show NEDD8 chain formation by mass spectrometry in these reactions. We find the fact that UBE2M also auto-modifies, something that we acknowledge in the paper, irrelevant in this context. This experiment is an

in vitro reaction, and as such, out of the cellular context, necessarily artificial, but it does prove the point that UBE2M is capable of making chains and that it is capable of making free chains. In addition to the *in vitro* experiment, we do provide cellular evidence of free chain formation. There is the simple fact that the neddylation pattern when examined by molecular mass nicely tracks the expected molecular masses of free chains (8, 16, 24 etc.). This pattern is not different in extract from NEDP1 knockout cells directly lysed in SDS buffer (ie preventing any enzymatic activity that could potentially cleave NEDD8 from a substrate after cell lysis) or in extract made under more gentle conditions that would maintain enzymatic activity. Furthermore, our mass spectrometry analysis is performed after in gel digestion of cut out bands. The fact that when we cut the smallest possible free chain (di-NEDD8) and subject that band to mass spec, we only find di-Gly motifs on NEDD8, and no other protein, strongly supports the presence of only a free di-NEDD8 chain in this slice. As under any circumstances, this band by molecular mass would otherwise have to correspond to a single NEDD8 moiety linked to another small protein target. The stoichiometry between that putative target and NEDD8 would have to be 1:1 and it should thus be equally abundant as NEDD8 and detectable. As a consequence, the identification of the di-Gy motif should be straight forward. Yet we only find NEDD8 and di-Glys on NEDD8 and this holds true for most of the higher molecular mass slices, strongly supporting the presence of free chains. Also, when we subject the purified tri-NEDD8 chain to 2D gel electrophoresis it behaves as a free tri-NEDD8 chain that, granted, is also post-translationally modified. We believe that all this data strongly points to the presence of free NEDD8 chains also in cells. The final point the referee makes is that we may just not detect the other substrate because the chain was cleaved during sample prep makes two assumptions: 1. That there is an as of yet undescribed NEDD8 specific protease in cells that cleaves NEDD8 chains from substrates, but leaves the chains intact and 2. That this enzyme is active during the lysis of cells in SDS sample buffer (as the neddylation pattern does not change between SDS lysed extract and extract lysed under milder conditions). While as above, it is inherently impossible to prove the absence of something, we believe that the cumulative evidence we provide with our data is more supportive of the presence of free unconjugated NEDD8 chains in cells than of this cryptic NEDD8 protease.

2.

The enrichment of PARP, while potentially there, is certainly not striking compared to the many other proteins that remain unlabeled.

We have updated the figure for our mass spectrometry results to emphasise more clearly why we focused on PARP-1. We have now labelled all known NEDD8 regulators in blue (protein that we would expect to co-purify with NEDD8) and PARP-1 and all its known substrates in red. We hope that this visual representation makes it more obvious why we decided to investigate PARP-1. We believe that this data, together with the obvious defect of PARP-1 activation in NEDP1 knockout cells, makes this decision an obvious choice.

3.

Fig. 3A: not clear why they find that the 25kD band is the most accumulating band in response to H₂O₂. Whether these conjugates indeed correspond to free NEDD8-trimers has not been tested.

We don't quite understand this comment. If one compares the Western blot prior to H₂O₂ treatment to after H₂O₂ treatment the 25kDa band is the one that is most strongly induced (we have indicated this band with an arrow). With this figure we don't claim that this band is free NEDD8 (this is done later with the 2D gel), but in Fig 3B we compare this H₂O₂ induced NEDD8 pattern to the pattern in non-stressed NEDP1 knockout cells, and find it to be identical, suggesting that these are the same conjugates.

4.

Fig. 3: the arguments that protection from H₂O₂-induced death in NEDP1 deleted cells is independent of the established CUL3-KEAP1 axis is indirect and not very convincing. It certainly does not provide strong evidence for a notion that free Nedd8-trimers mediate this effect.

In addition to showing that NRF2 is not significantly stabilised on the protein level in NEDP1 deleted cells, we have now also added qPCR data to show that transcription of NRF2 and its target NQO1 is not upregulated in NEDP1 deleted cells. We hope that this addresses this concern of the reviewer (Figure 3G)

5.

The phenotype of aberrant PAR formation in NEDP1-knockout cells is nice and indicates misregulation of PARP or related enzymes. This is also supported by their epistasis analysis.

We agree, and consider this important confirmation that we concentrated on the right proteins identified from our mass spec analysis.

6. Fig. 5C-G really require the whole NEDD8 blot - it is incomplete, and potentially very misleading, to only crop out the 25kD band. Fig.5D is overprocessed, suggesting that this interaction with this particular band is very unstable. As shown, these experiments cannot be interpreted at all, yet they would be an essential point for the authors. Frankly, I do not understand why they cut out the 25kD band, as it does suggest that other NEDD species bind and the mechanism of regulation might not be as straightforward as suggested by the authors. In addition, **inputs** and IPs have to be on the same gel - the current way of depicting the experiment is misleading and inappropriate.

We initially provided cropped gels to make the results more accessible, but given this concern have now updated all figures to show the entire molecular mass range of the gel.

7. The argument that Nedd8-binding to PARP inhibits the DNA-binding activity of the latter is very weak. To make such a broad statement, the authors would need to show that a significant fraction of PARP is associating with NEDD8 - which is not supported by the preliminary interaction studies shown in Fig. 5. They would also need to show directly that such competition takes place - binding to an overlapping domain, in the absence of any known Kd values, does not constitute strong evidence.

This is a hypothesis we postulate based on our results that PARP-1 activity is attenuated in NEDP1 knockout cells and that NEDD8 binds to the DNA binding domain of PARP-1. We have now included some more evidence that there is competitive binding between NEDD8 and DNA as we show that prior MNase treatment of extract (ie complete digestion of DNA) leads to stronger binding of tri-NEDD8 to PARP-1 (Fig S4E). The most straight forward explanation for an inhibition of PARP-1 by NEDD8 would be that DNA and NEDD8 compete for the same binding site, but we agree that this type of regulation could also happen in a more complicated manner. The exact mechanism of how this occurs will need to be the subject of further studies, but either way would not change the interpretation of the current study. We don't agree with the statement that a significant fraction of PARP-1 needs to be associated with NEDD8, as it only needs to be the fraction that is targeted to sites of DNA damage and hence "activated". This may very well be only a small fraction of the overall PARP-1 pool. We have now also added some text to suggest other regulatory mechanisms than direct competition between NEDD8 and DNA for PARP-1 binding.

8.

The decrease in binding to 25kD-NEDD8 (again, full blot needs to be shown) by HDAC-expression and the apparent increase by HDAC inhibition are minor and thus, evidence for a role of acetylation in regulating this interaction is weak. The potential role of acetylation in this pathway has not been addressed.

Full blots are now included. We believe the effect are clearly visible, in particular after overexpression of HDACs. We agree that NaB treatment only increases binding by approx. 2-fold, but as we see from the 2D-gels a significant proportion of tri-NEDD8 is already fully acetylated, so it is hard to imagine a very strong effect after NaB treatment. Nevertheless, it is measurable. Furthermore, these results need to be considered with the data in Fig 6F, where we show that HDAC overexpression rescues PARP-1 activity in NEDP1 knockout cells, clearly

demonstrating that acetylation is involved in PARP-1 regulation (also see response to referee 1 above).

9.

The last figure does not add much to the paper, and certainly does not provide a molecular mechanism for how oxidative stress in cells might trigger the proposed pathway.

This figure demonstrates that NEDP1 activity is much more sensitive to oxidation than UAE/UBE2M. This is an in vitro reaction and we were startled by how resistant UAE/UBE2M was to H₂O₂ levels, given that both UAE and UBE2M also contain active site cysteines that should be oxidized (just like the active site cysteine of NEDP1). We're thus somewhat puzzled by this comment, and while we agree that future work needs to show direct oxidation of the NEDP1 cysteine in cells, we believe that this in vitro data establishes that the forward reaction mediated by UAE/UBE2M is much less sensitive to oxidation than the reverse reaction mediated by NEDP1, and as such provides a hypothesis to test for how the effect we observe could occur in a cellular setting. We were very surprised by this result and thus have to strongly disagree with the statement of the referee.

2nd Editorial Decision

21st December 2018

Thank you for submitting your revised manuscript, which I now finally had a chance to carefully consider. I am happy to let you know that I found all remaining scientific issues well-addressed and responded to, and that we are therefore in principle ready to accept the manuscript for publication, following incorporation of a number of outstanding editorial issues.

Corresponding Author Name: Thimo Kurz
Journal Submitted to: EMBO Journal
Manuscript Number: EMBOJ-2018-100024R